# Provable Privacy Attacks on Trained Shallow Networks

## Abstract

We study what provable privacy attacks can be shown for trained 2-layer ReLU neural networks, focusing on two types of attacks: membership inference and data reconstruction. We prove that theoretical results on the implicit bias of 2-layer neural networks can be used to provably identify with high probability whether a given point was used in the training set in a high-dimensional, nearly orthogonal setting, and can also be used to construct a finite set of which at least a constant fraction are training points in a univariate setting. To the best of our knowledge, our work is the first to show provable vulnerabilities in this implicit-bias-driven setting.

## 1 Introduction

It is known that under certain conditions, trained neural networks do not merely interpolate their training data, but rather converge to specific solutions that also satisfy additional properties. This phenomenon, known as *implicit bias*, has been widely studied to better understand the generalization properties of neural networks. However, recent work showed that the implicit bias of training algorithms may also facilitate information leakage, by demonstrating that it can be exploited to reconstruct parts of the dataset used to train neural networks (Haim et al., 2022), or to infer with high probability whether a given instance was used in the training process or not (Golbari et al., 2025). Subsequent studies extended these techniques to broader settings (Buzaglo et al., 2023a;b; Andrew et al., 2023; Ye et al., 2023; Boenisch et al., 2024; Oz et al., 2024), highlighting this vulnerability as a practical concern. However, despite their theoretical motivation, most works do not rigorously explain when such attacks are successful and to what extent.

A recent paper (Refael et al., 2025) studied the reconstruction attack in Haim et al. (2022), demonstrating its brittleness in the absence of sufficient prior knowledge on the training data. While the former establishes the limitations of such implicit-bias-driven attacks, it nevertheless falls short of characterizing sufficient conditions under which such attacks are successful, and the resulting information leakage.

In this paper, we take a step at bridging this gap, and show what is to the best of our knowledge, the first provably successful privacy vulnerabilities induced by the above implicit bias, by rigorously showing that such attacks can be executed on trained neural networks under various assumptions. We show that under the assumptions that we identify, *all* neural networks that satisfy these implicit bias constraints must store at least some information on the training data, which can be used by a malicious attacker. More specifically, we use known results on the implicit bias of ReLU neural networks, which establish that such networks tend to converge to a certain margin maximization solution (Lyu and Li, 2020; Ji and Telgarsky, 2020). This characterization of the implicit bias of neural networks allows us to rigorously analyze certain cases in which the neural network memorizes the training data. In particular, this includes examples where an attacker is capable of reconstructing certain portions of the data in a univariate setting, or perform membership inference attacks with high success rates in a high dimensional, nearly orthogonal setting, effectively distinguishing between instances that are in the training set and fresh instances that were generated by the same distribution that was used to generate the training set.

While our attacks are applicable under certain input dimensions, we also conduct experiments demonstrating that these vulnerabilities can pose a broader concern, even when our assumptions on the dimension of the input are not met.

The remainder of the paper is structured as follows: after specifying our contributions in more detail below, we turn to discuss related work. In Section 2 we present our notations, some required background, and the main assumptions we make throughout the paper. In Section 3, we study membership inference attacks in high-dimensional settings. In Section 3.2 presents experiments that empirically support our findings, including scenarios in which our assumptions do not hold. Lastly, in Section 4 we study data reconstruction in the univariate setting.

**Our contributions**

Our main contribution is to provide rigorous guarantees in this implicit-bias-driven setting, as to the best of our knowledge, all previous work that studied this setting is empirical. In more detail, our contributions can be summarized as follows:

- We show that in the high-dimensional setting, under Assumption 3.1, which posits that training vectors are nearly orthogonal with high probability, an attacker can execute a membership inference attack with high success rates against architectures satisfying the implicit bias constraints (i.e., stationary points of the maximum-margin problem; see Assumption 2.1). Furthermore, we demonstrate that this near-orthogonality requirement is satisfied by several standard continuous distributions. In Subsection 3.1, we provide examples of various attacks, categorized by the level of information available to the adversary.

- We provide empirical evidence that the predicted membership-inference behavior persists beyond our theoretical setting, including in more moderate dimensions, with deeper architectures, and on MNIST; see Section 3.2 and Appendix E.

- Finally, we prove that in the univariate setting, under Assumption 2.1, which posits that the network weights converge to a stationary point of a maximum-margin problem, and two additional conditions stated in Theorem 4.4: that this stationary point is a local minimum of the constrained problem and that the network contains a neuron that is active on every training instance, an adversary can construct a finite set in which at least a constant fraction of the points are guaranteed to belong to the training set. Notably, this fraction is independent of both the training-set size and the network width. We present the reconstruction procedure in Algorithm 1.

**Related Work**

Privacy attacks in neural networks have been extensively studied in recent years. Since this paper focuses on two specific types of attacks, here we only review papers that study these kinds of attacks, or those that closely relate to them.

**Membership inference attacks.** Membership inference attacks (Shokri et al., 2017; Hu et al., 2022a; Olatunji et al., 2021; Shejwalkar et al., 2021) aim to determine whether a specific data point was part of the training set. Many of these attacks exploit the tendency of machine learning models to make more confident predictions on training data than on new, test data. Olatunji et al. (2021) applied this confidence-based technique to graph neural networks. Jha et al. (2020); Farokhi and Kaafar (2020) used information-theoretic tools to upper bound the success probability of membership inference attacks, whereas our work establishes provable lower bounds. Attias et al. (2024) also derive such attacks but focus on models with convex objective functions. Carlini et al. (2022); Zarifzadeh et al. (2024) introduced SOTA attacks that are based on the model's confidence of its prediction. Golbari et al. (2025) show a membership inference attack based on the implicit bias of trained neural networks, providing empirical evidence for the success of their method. Unlike their work, ours focuses on obtaining provable guarantees.

**Data reconstruction attacks.** The second type of attacks considered in this paper are data reconstruction attacks, which aim to fully recover the training set or parts of it. These include attacks on generative models such as large language models (Carlini et al., 2019; 2021; Nasr et al., 2023), diffusion models (Somepalli et al., 2022; Carlini et al., 2023), and in federated learning settings (Zhu et al., 2019; He et al., 2019; Hitaj et al., 2017; Geiping et al., 2020; Huang et al., 2021; Wen et al., 2022). Perhaps the most relevant works on reconstruction attacks are Haim et al. (2022); Buzaglo et al. (2023a); Oz et al. (2024). Using a known result on the implicit bias of trained neural networks, they define and optimize over a loss function, which upon empirical minimization, allows for the recovery of some of the training set. Inspired by these works, we use the same implicit-bias constraints and, under the additional local-optimality and always-active-neuron assumptions stated in Theorem 4.4, rigorously *prove* the existence of privacy vulnerabilities rather than merely demonstrate them empirically. While Refael et al. (2025) analyze the limitations of the Haim et al. (2022) attack, arguing that indistinguishable minima can lead to incorrect recovery, they offer no provable defenses. Specifically, they do not preclude an adversary from extracting requisite prior knowledge directly from the training set. Furthermore, unlike our work, they provide no provable guarantees for the attack itself, under any assumptions on the adversary's prior knowledge.

**Differential privacy.** While Differential Privacy (DP) (Dwork et al., 2006) provides a rigorous framework for bounding individual information exposure, our work adopts a different, complementary perspective. Given the empirical success of reconstruction attacks in standard training settings (Haim et al., 2022), it is evident that the implicit bias of neural networks often leads to non-negligible data leakage. Rather than establishing DP guarantees, we provide an alternative lens by formally quantifying the extent of information extraction possible within these inherently non-private regimes.

**Benign overfitting.** A related phenomenon in deep learning theory that may contribute to privacy vulnerabilities is benign overfitting (Bartlett et al., 2020; Cao et al., 2022; Li et al., 2021). Here, a neural network achieves perfect training accuracy while still generalizing well to unseen data, indicating that even high-performing models can memorize their training sets and become susceptible to privacy attacks. Although this offers a theoretical explanation, it does not directly provide or prove a method for extracting training set information as our work does.

## 2 Preliminaries and notation

In this section, we introduce the notations and settings used throughout this paper and discuss relevant background.

We consider a binary classification setting, where each data instance consists of a pair $(\mathbf{x}, y) \in \mathbb{R}^d \times \{-1, 1\}$, and we define the training set as $\{(\mathbf{x}_i, y_i)\}_{i=1}^n$ which consists of $n$ data points. We let $\Phi(\boldsymbol{\theta}; \cdot) : \mathbb{R}^d \to \mathbb{R}$ denote a neural network, where $\boldsymbol{\theta} \in \mathbb{R}^k$ are the parameters of the network represented as a vector. Let $\ell : \mathbb{R} \to \mathbb{R}$ denote the exponential loss function $z \mapsto e^{-z}$ or the logistic loss function $z \mapsto \log(1 + e^{-z})$, and let $L(\boldsymbol{\theta}) := \frac{1}{n} \sum_{i=1}^n \ell(y_i \cdot \Phi(\boldsymbol{\theta}; \mathbf{x}_i))$ be the empirical (training) loss. A network $\Phi(\boldsymbol{\theta}; \mathbf{x})$ is called *homogeneous* if there exists $c > 0$ such that for every $b > 0$, $\boldsymbol{\theta}$ and $\mathbf{x}$, it holds that $\Phi(b \cdot \boldsymbol{\theta}; \mathbf{x}) = b^c \Phi(\boldsymbol{\theta}; \mathbf{x})$. We use the shorthand $[x]_+ := \max(0, x)$ for the ReLU activation, and thus a homogeneous 2-layer ReLU network has the form $\Phi(\boldsymbol{\theta}, \mathbf{x}) = \sum_{j=1}^k v_j \left[ \mathbf{w}_j^\top \mathbf{x} + b_j \right]_+$ where $\boldsymbol{\theta}$ encapsulates the parameters $\{\mathbf{w}_j, v_j, b_j\}_{j=1}^k$. We denote the $(d-1)$-dimensional unit sphere in $\mathbb{R}^d$ by $\mathbb{S}^{d-1} := \{\mathbf{x} \in \mathbb{R}^d : \|\mathbf{x}\|_2 = 1\}$. We use standard asymptotic notation (e.g. $O, o, \Omega$, etc.).

The following known result characterizes the implicit bias in homogeneous neural networks by showing that these networks converge to a critical point of a certain margin-maximization problem.

**Theorem 2.1** (paraphrased version of Lyu and Li (2020), Ji and Telgarsky (2020)). *Let $\Phi(\boldsymbol{\theta}; x)$ be a homogeneous ReLU neural network. Consider minimizing the logistic $(z \mapsto \log(1 + e^{-z}))$ or the exponential $(z \mapsto e^{-z})$ loss using gradient flow (which is a continuous time analog of gradient descent) over a binary classification set $\{(x_i, y_i)\}_{i=1}^n \subseteq \mathbb{R}^d \times \{-1, 1\}$. Assume that there is a time $t_0$ where $L(\boldsymbol{\theta}(t_0)) < \frac{1}{n}$. Then, gradient*

*flow converges in direction[1] to a first order stationary point (KKT point) of the following maximum-margin problem:*

$$\min_{\boldsymbol{\theta}} \frac{1}{2}\|\boldsymbol{\theta}\|^2 \quad s.t \quad \forall i \in [n] \quad y_i \Phi(\boldsymbol{\theta}; x_i) \geq 1. \tag{1}$$

In light of the above result, it is natural to study the privacy implications under the assumption that our network has converged in direction to a KKT point of the above maximum-margin problem.[2] This implies a series of constraints on its weights, that are captured in the following assumption we make throughout the paper:

**Assumption 2.1.** *Let $\Phi(\boldsymbol{\theta}; \mathbf{x})$ be a 2-layer neural network, and let $m := \min_i |\Phi(\boldsymbol{\theta}; \mathbf{x}_i)| > 0$. We are given access to $\Phi(\boldsymbol{\theta}, \cdot)$, and we have full knowledge of the vector $\boldsymbol{\theta}$.[3] Moreover, we have that $\boldsymbol{\theta}$ satisfies the following KKT conditions of Eq. (1):*

$$\boldsymbol{\theta} = \sum_{i=1}^{n} \lambda_i y_i \nabla_{\boldsymbol{\theta}} \Phi(\boldsymbol{\theta}; \mathbf{x}_i), \tag{2}$$

$$\forall i \in [n], \quad y_i \Phi(\boldsymbol{\theta}; \mathbf{x}_i) \geq m > 0, \tag{3}$$

$$\lambda_1, \ldots, \lambda_n \geq 0, \tag{4}$$

$$\forall i \in [n], \quad if \ y_i \Phi(\boldsymbol{\theta}; \mathbf{x}_i) \neq m \ then \ \lambda_i = 0. \tag{5}$$

Since modern neural networks typically possess the power to perfectly interpolate the data or even random noise (Zhang et al., 2016), it is reasonable to assume that the trained network perfectly classifies the entire dataset, and that the requirements of Thm. 2.1 are thus satisfied. While the network might fail to achieve this in practice, we note that understanding potential privacy vulnerabilities is relevant mainly in cases where optimization has succeeded and obtained a useful network that exhibits good performance. Hence, we believe that this assumption on the success of the training process is natural when studying privacy. Moreover, in the literature on implicit bias, it is common to make this assumption, and to explore properties of the trained network in this case (Vardi, 2022).

We refer to the parameter $m$ as *the margin value*, and we say that a set of points $A \subseteq \mathbb{R}^d$ *lies on the margin* if $\Phi(\boldsymbol{\theta}; \mathbf{x})$ equals the margin value for all $\mathbf{x} \in A$. We stress that, in general, the attacker does not have knowledge of the value of $m$. Nonetheless, it is still possible that the attacker might be able to either deduce this value or obtain it in some way, and even if they cannot, this merely results in a single additional hyperparameter that the attacker must account for. For example, if the attacker knows the hyperparameters used for training (such as the learning rate and number of iterations), as commonly assumed in membership inference attacks (Carlini et al., 2022; Zarifzadeh et al., 2024), then they can approximately recover the margin by training reference models, which is a standard method in membership inference attacks. Moreover, even without any assumptions, the attacker can infer bounds on this value that are likely to hold in practical settings (see Remark 3.6), further indicating that our proposed attacks can reveal unwanted information. Throughout this paper, we present several results that vary based on the information we have on $m$.

## 3 High dimensional input

In this section, we focus on membership inference attacks. Given a point $\mathbf{x} \in \mathbb{R}^d$ that is either selected from the realized training set or sampled independently from the underlying distribution after training, the adversary aims to determine, with high probability, how $\mathbf{x}$ was generated.

In high dimensional settings, under many commonly used data distributions, we have that the dataset is nearly orthogonal with high probability. We exploit this property to show that also with high probability

---

[1]We say that gradient flow *converges in direction* to $\hat{\boldsymbol{\theta}}$ if $\lim_{t\to\infty} \frac{\boldsymbol{\theta}(t)}{\|\boldsymbol{\theta}(t)\|} = \frac{\hat{\boldsymbol{\theta}}}{\|\hat{\boldsymbol{\theta}}\|}$.

[2]Formally, this means that we assume that the parameter vector $\boldsymbol{\theta}$ of the network satisfies $\boldsymbol{\theta} = \gamma\hat{\boldsymbol{\theta}}$, where $\hat{\boldsymbol{\theta}}$ is a KKT point and $\gamma > 0$ is some scalar.

[3]Many of our results or similar ones can be proven even with only partial access to the network's weights, however for the sake of simplicity we assume full knowledge of the weights.

over drawing the training set, all the points in the training set lie on the margin. On the other hand, if we were to draw a new data point from the same distribution, the neural network would produce a target value that is typically much smaller than the margin. These key observations will allow us to make the distinction between training points and test points, effectively answering membership inference queries.

**Remark 3.1** (Black box attacks). *We note that since our results in this section are only based on querying the value of $\Phi(\boldsymbol{\theta}; \cdot)$, the attacker need not know $\boldsymbol{\theta}$ to successfully execute our proposed membership inference attack, and therefore the attack can also be applied in the black box model.*

Next, we now formally state our assumptions on the underlying distribution $\mathcal{D}$ which generates the dataset.

**Assumption 3.1.** *The following holds for some $\tau > 0$.*

1. *For $\mathbf{x}_1, \mathbf{x}_2 \sim \mathcal{D}, \quad \Pr[n \cdot |\mathbf{x}_1^\top \mathbf{x}_2| \leq o(d)] \geq 1 - \frac{\tau}{n^2}.$*

2. *For $\mathbf{x} \sim \mathcal{D}, \quad \Pr\left[\|\mathbf{x}\|^2 \geq \Omega(d)\right] \geq 1 - \frac{\tau}{n}.$*

*where $n$ is the size of the training set.*

We remark that our results do not depend on the data labels and thus hold for any arbitrary labeling scheme. We also point out that even though this assumption may seem somewhat restrictive at a first glance, it can be expected to hold for continuous distributions in sufficiently large dimensions, and when the sample size is modest. We also prove that Assumption 3.1 is satisfied by several rather standard data distributions. This includes (but is not limited to) the following concrete examples:

1. The uniform distribution over the sphere $\sqrt{d} \cdot \mathbb{S}^{d-1}$, where $n = o\left(\frac{\sqrt{d}}{\log d}\right)$ and $\tau = o_d(1)$.

2. The normal distribution $\mathcal{N}(\boldsymbol{\mu}, I)$ with mean $\boldsymbol{\mu}$, where $\|\boldsymbol{\mu}\|^2 = o(d)$, and where $n = \frac{o(d)}{\|\boldsymbol{\mu}\|^2 + d^\epsilon}$ for some $\frac{1}{2} < \epsilon < 1$ and $\tau = o_d(1)$.

3. Mixture of $k$ Gaussians with means $\boldsymbol{\mu}^{(1)}, \ldots, \boldsymbol{\mu}^{(k)}$, where $\|\boldsymbol{\mu}^{(1)}\|^2, \ldots, \|\boldsymbol{\mu}^{(k)}\|^2 = o(d)$, identity covariance matrices, $n = \frac{o(d)}{\max\{\|\boldsymbol{\mu}^{(i)}\|^2\}_{i=1}^k + d^\epsilon}$ for some $\frac{1}{2} < \epsilon < 1$, and $\tau = o_d(1)$.

The first two examples are rather standard in the literature, whereas the last example is somewhat more complex, but is meant to exemplify a setting where our proposed attacks can be executed in the statistically learnable case. For a more formal discussion about the statistically learnable case, we refer the reader to Appendix C. For proofs that these distributions satisfy Assumption 3.1, we refer the reader to Appendix D.

Before we continue, we will introduce some further notation to be used throughout this section. Recall that $m > 0$ denotes the value of the network's margin, and define $\delta := \max_{i \neq j}\left\{|\mathbf{x}_i^\top \mathbf{x}_j|\right\}$ and $\Delta := \min_{i \in [n]}\left\{\|x_i\|^2\right\}$. Note that by Assumption 3.1 and by the union bound, we have that $n \cdot \delta = o(\Delta)$ with probability at least $1 - 2\tau$.

Given a point $\mathbf{x} \in \mathbb{R}^d$, we would like to infer whether $\mathbf{x}$ was in the training set, or if it was generated from the same distribution that generated the training set. As previously discussed, our strategy is to calculate the value of $|\Phi(\boldsymbol{\theta}; \mathbf{x})|$. We expect to see larger values that are closer to the margin when $\mathbf{x}$ is in the training set, and smaller values when it is not. Formalizing this idea, the following theorem is used to determine w.h.p. whether a given point $\mathbf{x} \in \mathbb{R}^d$ is in fact a training point, or a test point which was freshly sampled from $\mathcal{D}$.

**Theorem 3.2.** *Let $\mathcal{D}$ be a distribution on $\mathbb{R}^d$ that satisfies Assumption 3.1. Let $\mathbf{x} \in \mathbb{R}^d$ and let $\Phi(\boldsymbol{\theta}; \cdot)$ be a 2-layer neural network satisfying Assumption 2.1. Then the following hold:*

- *With probability at least $1 - 2\tau$ over the choice of the training set, if $\mathbf{x}$ is a training point then $|\Phi(\boldsymbol{\theta}; \mathbf{x})| = m$.*

- *If $\mathbf{x} \sim \mathcal{D}$ is sampled independently of the training set, then with probability $1 - 4\tau$ over the sampling of $\mathbf{x}$ and the training set, $|\Phi(\boldsymbol{\theta}; \mathbf{x})| = O\left(\frac{n \cdot m \cdot \delta}{\Delta}\right) = o_d(m)$.*

This theorem gives us a useful tool for performing membership inference attacks. Given a point $\mathbf{x} \in \mathbb{R}^d$, run $\mathbf{x}$ through the neural network, and observe whether $|\Phi(\boldsymbol{\theta}; \mathbf{x})| = m$ or $|\Phi(\boldsymbol{\theta}; \mathbf{x})|$ is much smaller than $m$.

We note that popular membership-inference attacks, such as Carlini et al. (2022) and Zarifzadeh et al. (2024), rely on the observation that a trained model tends to have larger outputs (or equivalently, smaller loss, or predictions with a higher confidence) on training examples compared to fresh test examples. Thus, Thm. 3.2 shows that this empirical observation provably holds for 2-layer networks with high-dimensional data.

The intuition behind the proof of the theorem can be explained as follows: Using Assumption 2.1, we show that the value $|\Phi(\boldsymbol{\theta}; \mathbf{x})|$ can be expressed as a weighted combination of $\{\mathbf{x}_i^\top \mathbf{x}\}_{i=1}^n$, where $\{\mathbf{x}_i\}_{i=1}^n$ are the training points. By Assumption 3.1, we have that if $\mathbf{x}$ is in the training set, then $\mathbf{x} = \mathbf{x}_k$ for some $k \in [n]$ and $\|\mathbf{x}_k\|^2$ must be large, while $|\mathbf{x}_j^\top \mathbf{x}_k|$ is small for all $j \neq k$. The main challenge in the proof is that the combination of all $|\mathbf{x}_j^\top \mathbf{x}_k|$ can be large and overwhelm the single quantity $\|\mathbf{x}_k\|^2$. By a careful analysis of the KKT constraints, we show that this is not the case, as that the weighted combination is "approximately balanced". On the other hand, when $\mathbf{x} \sim \mathcal{D}$, then with high probability it is "nearly orthogonal" to all training points, implying that $|\mathbf{x}_j^\top \mathbf{x}|$ is small for all $j = 1, \ldots, n$. Likewise, a careful analysis reveals that this weighted combination is also balanced, and is therefore small. For the complete proof, we refer the reader to Appendix B.

## 3.1 Example use cases of Thm. 3.2

Having presented our main tool in this section, we now turn to discuss several particular use cases, based on the amount of knowledge known to the attacker. We first assume that the value of the margin is known to the attacker. However, since an attacker cannot deduce the value of the margin in general, we also provide examples where membership inference questions can be answered without this knowledge.

In the following analysis, let $\Phi(\boldsymbol{\theta}; \mathbf{x})$ be a 2-layer neural network satisfying Assumption 2.1 and $\mathcal{D}$ be a distribution satisfying Assumption 3.1, thereby fulfilling the conditions of Thm. 3.2.

**Corollary 3.3** (Known margin value). *Let $\mathbf{x} \in \mathbb{R}^d$, assume that $d$ is sufficiently large, and further assume that we know the value of the margin $m$. Then, w.h.p. over the randomness in sampling the training set from $\mathcal{D}$, we have that:*

- *If $\mathbf{x}$ is in the training set then $|\Phi(\boldsymbol{\theta}; \mathbf{x})| = m$.*
- *If $\mathbf{x} \sim \mathcal{D}$ is a fresh example, then w.h.p. over the randomness in sampling $\mathbf{x}$, $|\Phi(\boldsymbol{\theta}; \mathbf{x})| < \frac{m}{2}$.*

*Proof.* From Thm. 3.2, we know that w.h.p. over the choice of the training set, we have that if $\mathbf{x}$ is in the training set then $|\Phi(\boldsymbol{\theta}; \mathbf{x})| = m$, and if $\mathbf{x} \sim \mathcal{D}$ then w.h.p. $|\Phi(\boldsymbol{\theta}; \mathbf{x})| \leq O\left(\frac{n \cdot m \cdot \delta}{\Delta}\right) = m \cdot O\left(\frac{n \cdot \delta}{\Delta}\right) < \frac{m}{2}$, where in the last inequality we used the fact that $O(\frac{n \cdot \delta}{\Delta}) = o_d(1)$, and the assumption that $d$ is sufficiently large. $\quad\square$

Thus, by the above, if the margin value $m$ is known to the attacker, they can simply compute $|\Phi(\boldsymbol{\theta}; \mathbf{x})|$ and return that $\mathbf{x}$ is in the training set if and only if $|\Phi(\boldsymbol{\theta}; \mathbf{x})| \approx m$.

As previously discussed, in general, the value of the margin is not known to the attacker. Nonetheless, under different assumptions, the attacker can still execute a successful membership inference attack.

**Corollary 3.4** (Leaked data point). *Let $k$ be a constant (independent of $d$), let $\mathbf{z}_1, \ldots, \mathbf{z}_k \sim \mathcal{D}$ be $k$ points, and assume we know that at least one point in this set is in the training set. Let $\alpha = \max_{1 \leq i \leq k} \{|\Phi(\boldsymbol{\theta}; \mathbf{z}_i)|\}$, then w.h.p. over the choice of the training set, we have for all $i \in [k]$:*

- *If $\mathbf{z}_i$ is in the training set then $|\Phi(\boldsymbol{\theta}; \mathbf{z}_i)| = \alpha$.*
- *If $\mathbf{z}_i \sim \mathcal{D}$ then w.h.p. (over sampling $\mathbf{z}_i$) $|\Phi(\boldsymbol{\theta}; \mathbf{z}_i)| < \frac{\alpha}{2}$.*

*Proof.* W.l.o.g. let $\mathbf{z}_1$ be in the training set. Using Thm. 3.2 and the union bound over $\mathbf{z}_1, \ldots, \mathbf{z}_k$, we have $|\Phi(\boldsymbol{\theta}, \mathbf{z}_i)| \leq m$ for all $i$ with probability at least $1 - 4k\tau = 1 - o_d(1)$, so in particular $\alpha \leq m$. On the

other hand, using Thm. 3.2 again, we have that with probability at least $1 - 2\tau = 1 - o_d(1)$ we have that $|\Phi(\boldsymbol{\theta}, \mathbf{z}_1)| = m$, so $m \leq \alpha$. Thus, w.h.p. $m = \alpha$. By using Corollary 3.3, the proof follows. □

This corollary implies that even without knowing the margin value, an attacker aware that at least one element of a set of size $k$ belongs to the training set can identify it as the element achieving the largest absolute prediction value within that set. This allows the attacker to deduce the margin value by computing $\max_i |\Phi(\boldsymbol{\theta}; \mathbf{z}_i)|$. Thereafter, the attacker can continue in the same manner as in Corollary 3.3.

One might argue that even the previous assumptions are somewhat restrictive, since they require that at least one training point is leaked a priori. The following corollary makes some additional assumptions on the underlying distribution and that the margin value is bounded rather than known, which is much milder than in the previous result.

**Corollary 3.5** (Bounded margin). *Let $\mathcal{D}$ be a distribution that satisfies the following slightly stronger version of Assumption 3.1:*

*Let $\tau > 0$.*

- *For $\mathbf{x}, \mathbf{y} \sim \mathcal{D}$, $n \cdot |\mathbf{x}^\top \mathbf{y}| = o\left(\frac{d}{t(d)}\right)$ for some function $t(d)$ with probability at least $1 - \frac{\tau}{n^2}$.*

- *For $\mathbf{x} \sim \mathcal{D}$, $\|\mathbf{x}\|^2 \geq \Omega(d)$ with probability at least $1 - \frac{\tau}{n}$.*

*Furthermore, let $\mathbf{x} \sim \mathcal{D}$ and suppose that $C < m < t(d)$ for some constant $C$. Then we have:*

- *W.p. at least $1 - 2\tau$ over the training set, if $\mathbf{x}$ is in the training set then $|\Phi(\boldsymbol{\theta}; \mathbf{x})| > C$.*

- *If $\mathbf{x} \sim \mathcal{D}$ then w.p. at least $1 - 4\tau$ over the training set and $\mathbf{x}$, $|\Phi(\boldsymbol{\theta}; \mathbf{x})| < o_d(1)$.*

*Proof.* Assume that $\mathbf{x}$ is in the training set. From Thm. 3.2 we know that $|\Phi(\boldsymbol{\theta}; \mathbf{x})| = m > C$ with probability at least $1 - 2\tau$. Assume that $\mathbf{x}$ is not in the training set. From Thm. 3.2 and our stronger assumption on $\mathcal{D}$ we know that

$$|\Phi(\boldsymbol{\theta}; \mathbf{x})| = O\left(\frac{n \cdot \delta \cdot m}{\Delta}\right) \leq O\left(o\left(\frac{d}{t(d)}\right) \cdot \frac{m}{\Delta}\right)$$
$$= O\left(o\left(\frac{d}{t(d)}\right) \cdot \frac{m}{d}\right) = o_d(1) ,$$

with probability at least $1 - 4\tau$. □

This corollary implies the following: for $\mathbf{x} \in \mathbb{R}^d$, consider the value of $|\Phi(\boldsymbol{\theta}; \mathbf{x})|$. If $\mathbf{x}$ is not in the training set, then w.h.p. we get a value which is smaller than $C$, and if $\mathbf{x}$ is in the training set, then w.h.p. we get a value which is greater than $C$.

**Remark 3.6** (On the lower and upper bounds of the margin). *We argue that the margin assumptions used in Corollary 3.5 are mild. With exponential or logistic loss functions (standard assumptions in this setting), the gradient becomes exponentially small as the margin grows. Thus, if the margin is even polylogarithmic in $d$, further training becomes highly inefficient. This suggests that we can choose $t(d) = \text{polylog}(d)$ to satisfy our assumption, which is therefore only slightly stronger than Assumption 3.1. Conversely, a very small margin implies large loss on margin points, suggesting that training stopped prematurely. For a more formal justification, see Remarks B.5 and B.6.*

### 3.2 An experiment for moderate values of $d$

Thus far, our theoretical guarantees require the input dimension to be sufficiently large for Assumption 3.1 to hold. The following experiments are not intended to validate these rigorous guarantees within their stated regime, but rather to explore whether the qualitative behavior they predict persists in more moderate dimensions, beyond the scope of our assumptions.

Studying this question empirically, in this section, we conduct a few simulations focusing on the membership inference problem, and observe that while our theoretical results' assumptions do not necessarily hold, their implications are nevertheless still valid. We sampled training and test sets (both i.i.d.) from a mixture of two Gaussian distributions, trained a 2-layer neural network until reaching an approximate KKT point, and examined the network's predictions on both the training and the test sets in comparison to the margin.

More specifically, we conducted all our simulations using the following settings:

- **Architecture:** We focused on 2-layer ReLU networks, where the hidden layer has 10,000 neurons. The neurons in the hidden layer each have a bias term while the second layer does not, thus making the network homogeneous.

- **Range of the input dimension:** We tested $d$ for various values in the range between 1 and 1,000. This range includes values of $d$ where it is much larger than the training set in our theoretical results, but also includes more moderate values of $d$ where our assumptions do not necessarily hold.

- **Data generation:** All points were sampled i.i.d. from a mixture of two Gaussians, with means $(\pm 1, 0, \ldots, 0) \in \mathbb{R}^d$ and identity covariance matrices. The training set contains 20 instances, since this small size ensures that Assumption 3.1 holds for the larger values of $d$ that we tested. The test set contains 5,000 instances.

- **Training:** In order to converge faster to an approximate KKT point, we used a small initialization scheme as was done in Haim et al. (2022). After training, the network achieved 100% training accuracy, and 85% or more test accuracy in all reported experiments.

Our experiment focused on studying two objectives. The first studies how many training points lie on the margin as a function of the dimension $d$,[4] and the second studies how many test points that were sampled from the same distribution as the training set lie on or above the margin.

Our results demonstrate that network outputs can serve as effective tools for privacy attacks across a broader range of input dimensions, suggesting wider applicability of our theory. Specifically, Fig. 1a shows that as input dimensions increase, more training points lie on the margin, indicating a higher probability of this occurrence. Similarly, Fig. 1b reveals that the number of test points lying on or above the margin decreases with higher dimensions, implying a reduced likelihood of test points from the same distribution doing so. Notably, these findings align with our theory and extend to much smaller dimensions than predicted. For instance, while Thm. 3.2 suggests a minimum dimension of $d \approx n^2 = 400$[5] for a training set of size 20, our experiments show that nearly all test points fall below the margin even at $d = 100$, and about 80% do so at $d = 20$, highlighting the potential for membership inference attacks at much smaller dimensions. All results in Figure 1 report relative values of training and test points compared to the margin, averaged over 10 instantiations.

Following our empirical findings, we conclude that our theory is expected to hold more generally, and that the magnitude of the output of the neural network on a data instance can reveal whether it is a training point or a test point with high success rates. This is in line with many empirical findings (see Hu et al. (2022b)), and provides a theoretical explanation for this phenomenon.

Appendix E further examines settings beyond our theoretical scope, including deeper architectures and MNIST.

## 4    One-dimensional input

We now turn to a more potent privacy threat: the reconstruction attack. In this setting, the adversary aims to recover either specific portions or the entirety of the training set. Since reconstruction is strictly more

---

[4]It is noteworthy that a similar experiment was conducted in Vardi et al. (2022b), albeit under a different context where the adversarial robustness of the neural network is studied.

[5]This is because of the fact that under the assumption $n = \sqrt{d}$, we have w.h.p. that $n \cdot |\mathbf{x}_1^\top \mathbf{x}_2| = \Theta(d)$, so Assumption 3.1 is very unlikely to hold for values of $d$ that are smaller than that.

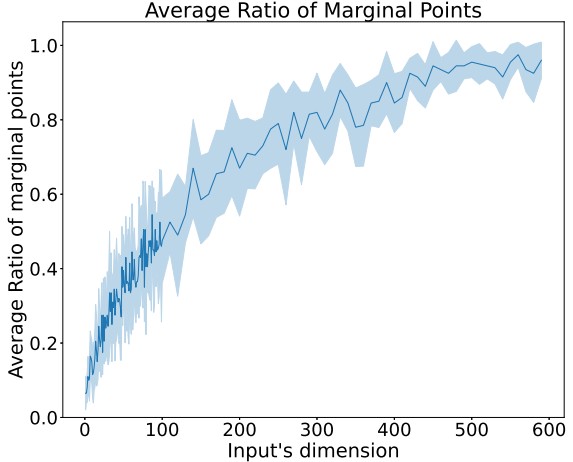 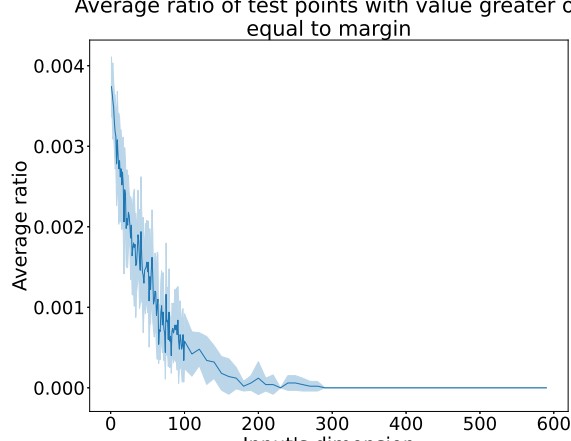

(a) The percentage of training points that lie on the margin (up to a slack of 10%) increases as the dimension increases.

(b) The percentage of test points that lie on or above the margin drops to zero for sufficiently large input dimensions, much earlier than what our theory predicts.

Figure 1: Results for the 2-layers fully connected architecture on Gaussian distribution as a function of the input dimension. As the dimension increases, a larger fraction of training points lie on the margin (a), while a smaller fraction of test points lie on or above the margin (b). Results are averaged over 10 independent runs.

challenging than membership inference, as recovering an instance inherently identifies it as a training member, it is natural to require stronger assumptions to guarantee success. Refael et al. (2025) demonstrated that for input dimensions $d \geq 2$, the reconstruction attack of Haim et al. (2022) is generally unreliable without significant prior knowledge. In this section, we analyze the efficacy of this attack in the univariate setting ($d = 1$). Under the additional assumptions stated in Theorem 4.4, namely local optimality and the existence of a neuron active on all training instances, we prove that an attacker can construct a finite candidate set of which at least a constant fraction are training samples. We also propose an alternative reconstruction procedure that avoids these assumptions, albeit with narrower applicability (see Appendix G).

We consider univariate ReLU networks of the form

$$x \mapsto \sum_{j=1}^{k} v_j \left[ w_j x + b_j \right]_+ , \tag{6}$$

where $x \in \mathbb{R}$. Such networks compute piecewise linear functions whose breakpoints (the 'kinks' where the slope changes) are given by $\{-\frac{b_j}{w_j}\}_{j=1}^k$. We assume without loss of generality that $-\frac{b_1}{w_1} < \ldots < -\frac{b_k}{w_k}$. Throughout this section, we further assume the attacker knows $m$.

### 4.1 Warming up – the case $n = k = 1$

To illustrate a scenario where full reconstruction is provably possible, we first consider the simplest case where $n = k = 1$. This pedagogical example highlights the fundamental geometric link between the network's breakpoint and the training data. Formally, we establish the following:

**Theorem 4.1.** *Suppose that $\Phi(\boldsymbol{\theta}; \cdot)$ is a univariate neural network as in Eq. (6), and that Assumption 2.1 holds. Moreover, suppose that $n = k = 1$. Then, there exists a single solution $x$. Furthermore, it can be easily recovered if $m$ is known.*

*Proof.* Assume by contradiction that $y_1 \Phi(\boldsymbol{\theta}; x_1) \neq m$. Then by Eq. (5), we have that Eq. (2) must equal zero, implying that $\Phi$ is the zero function, which contradicts Eq. (3). We thus deduce that $y_1 \Phi(\boldsymbol{\theta}; x_1) = m$.

Since $y_1 \in \{\pm 1\}$, we must have $\Phi(\boldsymbol{\theta}; x_1) \in \{\pm m\}$. By our assumption that $k = 1$, $\Phi$ takes the form $\Phi(\boldsymbol{\theta}; x) = v_1 [w_1 x + b_1]_+$. This function equals zero whenever the ReLU neuron is inactive, and is necessarily not zero (since it would contradict Eq. (3)) whenever the neuron is active, thus it has a nonzero slope and equals either $-m$ or $m$ at a unique point which is necessarily $x_1$. □

Although the above example is highly degenerate, it nevertheless highlights the danger and exemplifies the impact this theoretical tool may have in practice, and further motivates us to explore whether such vulnerabilities exist in more general settings.

## 4.2   The general univariate case

As we will see in this subsection, fully recovering the dataset in the general univariate case is significantly more complex than in the previous case, if possible at all. However, we show that under our assumptions, some instances of the training set can still be extracted.

Our previous analysis relied on the KKT conditions, which suggested that points on the margin are potential training points. However, it is unclear whether this holds in general or what fraction of such points actually belong to the training set. In the univariate case, the neural network may either cross the margin with a nonzero slope or remain flat along an interval at the margin. In the former case, at most two candidate points arise per linear interval where the network crosses the margin. In the latter, there is a continuum of candidates. However, a careful analysis shows that in both cases, there is a finite set of candidates which must contain a training point.

The following theorems address these cases and establish the existence of a discrete set guaranteed to contain a training point. All proofs can be found in Appendix F.

**Theorem 4.2.** *Let $\Phi(\boldsymbol{\theta}; x)$ be a 2-layer univariate network satisfying Assumption 2.1. Let $[-\frac{b_{i-1}}{w_{i-1}}, -\frac{b_i}{w_i}]$ and $[-\frac{b_i}{w_i}, -\frac{b_{i+1}}{w_{i+1}}]$ be two adjacent intervals which none of them is constant on the margin. Then, there must be a training point in the interval $[-\frac{b_{i-1}}{w_{i-1}}, -\frac{b_{i+1}}{w_{i+1}}]$, and this training point must lie on the margin. Moreover, the number of points lying on the margin in this interval is at most 4.*

The proof of the above theorem relies on the observation that for any three breakpoints, two of them must belong to neurons with the same sign of the parameter $w$. If these two neurons are active on the same set of training points, then by Assumption 2.1, they merge into a single neuron, therefore there must exist some training point between them. Moreover, this training point must lie on the margin. Since each interval crosses the margin at most twice, the number of possible points lying on the margin is at most four.

Having presented our theorem for the case where the neural network is not constant on the margin, we now present our theorem for the complementary case where it is constant.

**Theorem 4.3.** *Let $\Phi(\boldsymbol{\theta}; x)$ be a 2-layer univariate network satisfying Assumption 2.1. In addition, suppose that:*

- *There is a neuron $c_1$ that is active on all the points in the training set.*

- *$\Phi(\boldsymbol{\theta}; x)$ is a local optimum of Eq. (1).*

- *$\Phi(\boldsymbol{\theta}; x)$ alternatingly lies on the margin on three adjacent intervals, i.e. it is constant on $[-\frac{b_{i-2}}{w_{i-2}}, -\frac{b_{i-1}}{w_{i-1}}]$ and on $[-\frac{b_i}{w_i}, -\frac{b_{i+1}}{w_{i+1}}]$ (but not in between) and lies on the margin, for some $i$.*

*Then, either $-\frac{b_{i-1}}{w_{i-1}}$ or $-\frac{b_i}{w_i}$ is a training point.*

To prove this theorem, we observe that if by contradiction, neither $-\frac{b_{i-1}}{w_{i-1}}$ nor $-\frac{b_i}{w_i}$ is a training point, then we can construct a modified network with a slightly different breakpoint $-\frac{b_i}{w_i} + \epsilon$, for any $\epsilon > 0$. We show that this new network has strictly smaller norm, yet it is still a feasible solution of Eq. (1), contradicting $\Phi(\boldsymbol{\theta}, \cdot)$ having minimal norm.

We note that in terms of the structure of the function $\Phi(\boldsymbol{\theta}; \cdot)$, the above case analysis is exhaustive (excluding degenerate cases such as a network $\Phi(\boldsymbol{\theta}; \cdot)$ which consists of at most two different intervals, on which it is linear). This holds true since if the conditions in Thm. 4.3 do not hold, then this implies that $\Phi(\boldsymbol{\theta}; \cdot)$ does not lie on the margin in two adjacent intervals, hence the conditions for Thm. 4.2 must hold. We also remark that we have assumed that there is a neuron which is active on all the training data points, which typically makes sense in settings where the network is highly over-parameterized.

Combining our previous two theorems, we obtain the following.

**Theorem 4.4.** *Let $\Phi : \mathbb{R} \to \mathbb{R}$ be a 2-layer homogeneous network satisfying Assumption 2.1. In addition, assume:*

- *There is a neuron $c_1$ that is active on all the points in the training set.*

- *$\Phi(\boldsymbol{\theta}; x)$ is a local optimum of Eq. (1).*

*Then, Algorithm 1 constructs a finite set of which a constant fraction $p \geq \frac{1}{4}$ of the points are training points.*

---

**Algorithm 1:** Construct a finite set of candidates

$S \leftarrow \emptyset$
**for** $i \leftarrow 1$ **to** $k - 2$                                           // Iterating over break points
**do**

$\quad x \leftarrow -\frac{b_i}{w_i}$,
$\quad y \leftarrow -\frac{b_{i+1}}{w_{i+1}}$,
$\quad z \leftarrow -\frac{b_{i+2}}{w_{i+2}}$
$\quad$ **if** *both $[x, y]$ and $[y, z]$ do not lie on the margin* **then**
$\quad\quad S \leftarrow S \cup \{p : p \in [x, y] \wedge p$ is on the margin$\}$
$\quad\quad S \leftarrow S \cup \{p : p \in [y, z] \wedge p$ is on the margin$\}$
$\quad$ **if** *$[x, y]$ lies on the margin **and** $i < n - 2$* **then**
$\quad\quad t \leftarrow -\frac{b_{i+3}}{w_{i+3}}$
$\quad\quad$ **if** *$[z, t]$ lies on the margin* **then**
$\quad\quad\quad S \leftarrow S \cup \{y\} \cup \{z\}$

---

Algorithm 1 essentially iterates over the linear intervals of the network, and uses either Thm. 4.2 or Thm. 4.3 based on the structure of $\Phi(\boldsymbol{\theta}; \cdot)$ to add a constant number of candidate points, until the final set of points is constructed. We point out that we have assumed that $\boldsymbol{\theta}$ is a local optimum of Eq. (1) rather than just a critical point. It is known that in general, not all critical points of Eq. (1) are also local optima, and that gradient flow may converge to a critical point which is not a local optimum (see Safran et al. (2022, Example 1)), but it is not clear what is the 'typical' behavior of gradient flow in this context. We also remark that despite our requirement to have full knowledge of $\boldsymbol{\theta}$, the above results can also be implemented with partial knowledge of $\boldsymbol{\theta}$.[6] In any case, we leave the exploration of other privacy related questions on relaxations of our assumptions for future work.

## 5 Beyond Exact KKT

Our membership-inference guarantee is stable under small Euclidean perturbations of an exact KKT solution. Let $\boldsymbol{\theta}^*$ be a KKT point to which Theorem 3.2 applies, and suppose that $\|\boldsymbol{\theta} - \boldsymbol{\theta}^*\|_2 \leq \epsilon$. For every fixed input

---

[6]For example, if we have access to $\Phi(\boldsymbol{\theta}; \cdot)$ and only the breakpoints where the network changes its linearity are known, we can still interpolate and compute the points which cross the margin.

$\mathbf{x}$, the map $\boldsymbol{\theta} \mapsto \Phi(\boldsymbol{\theta}; \mathbf{x})$ is locally Lipschitz, and hence

$$\left| \Phi(\boldsymbol{\theta}; \mathbf{x}) - \Phi(\boldsymbol{\theta}^*; \mathbf{x}) \right| \leq L(\mathbf{x})\epsilon,$$

where $L(\mathbf{x}) = O\big((\|\boldsymbol{\theta}^*\|)(1 + \|\mathbf{x}\|)\big)$. Consequently, a training point satisfies $|\Phi(\boldsymbol{\theta}; \mathbf{x})| \geq m - L(\mathbf{x})\epsilon$, whereas a fresh point satisfies $|\Phi(\boldsymbol{\theta}; \mathbf{x})| \leq o_d(m) + L(\mathbf{x})\epsilon$. Thus, the two regimes remain separated whenever $L(\mathbf{x})\epsilon = o_d(m)$.

The univariate reconstruction guarantee admits a similar stability interpretation. Let $\boldsymbol{\theta}^*$ be an exact KKT point satisfying the assumptions of Section 4, and suppose that $\|\boldsymbol{\theta} - \boldsymbol{\theta}^*\|_2 \leq \epsilon$. The candidates constructed by Algorithm 1 are determined by the network's breakpoints and by its intersections with the margin. Both vary continuously with the parameters. More precisely, provided that the relevant neuron weights and the slopes of the affine pieces are bounded away from zero (similarly to the setting studied in Algorithm 2 in the appendix), a perturbation of size $\epsilon$ changes each candidate location by at most $r(\epsilon) = O(\epsilon)$, with a constant depending inversely on these non-degeneracy bounds. The lower bound on the slopes is essential: it converts a small perturbation of the network output into a small displacement of its intersection with the margin.

Accordingly, applying the reconstruction procedure to $\boldsymbol{\theta}$ yields a perturbed version of the candidate set obtained from $\boldsymbol{\theta}^*$. Replacing each reconstructed candidate $\hat{x}$ by the interval

$$[\hat{x} - r(\epsilon), \ \hat{x} + r(\epsilon)]$$

therefore produces a collection of intervals of which at least the same constant fraction $p \geq \frac{1}{4}$ contain a training point. As $\epsilon \to 0$, these intervals collapse to the finite candidate set of Theorem 4.4.

## 6 Summary and discussion

In this paper, we provide what is, to our knowledge, the first rigorous analysis of sufficient conditions for implicit-bias-driven privacy attacks. A tantalizing open problem remains to further relax our assumptions, which would deepen our understanding of the scope of these vulnerabilities and the conditions under which they might be circumvented. While we leave the design of provable defenses to future work, we hope our results motivate further rigorous study of the privacy-utility trade-offs inherent in the implicit bias of trained neural networks.

**Limitations and future work.** Our analysis is limited to 2-layer ReLU networks. While the implicit-bias characterization we rely on extends to arbitrary-depth homogeneous networks, exploiting it to obtain a tractable description of the memorized data becomes substantially more challenging beyond two layers, and remains an important open problem. Furthermore, our reconstruction guarantees are restricted to the univariate setting. In higher dimensions, reconstruction is known to be brittle Refael et al. (2025), motivating our focus on the weaker but more robust task of membership inference. Finally, our high-dimensional results rely on the near-orthogonality assumption (Assumption 3.1), which we adopt as a tractable sufficient condition for data separation. Since $\Phi(\boldsymbol{\theta}; \cdot)$ is Lipschitz, nearby points produce similar outputs, suggesting that some form of separation is needed for our output-based membership-inference argument. Near-orthogonality provides a convenient condition for establishing this separation and holds with high probability for several standard high-dimensional distributions considered here.

**Broader impact statement.** Privacy attacks raise dual-use concerns, but studying them can reveal when trained models leak information and inform privacy evaluation, safer training procedures, and defenses. These results are intended for controlled, authorized use to assess and reduce privacy risks.

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

## A  Notations

Denote by $\sigma_j'$ the subgradient of $\left[\mathbf{w}_j^\top \mathbf{x} + b_j\right]_+$. If $\mathbf{w}_j^\top \mathbf{x} + b_j \neq 0$ then $\sigma_j'$ is well defined, and if $\mathbf{w}_j^\top \mathbf{x} + b_j = 0$ then $\sigma_j' \in [0, 1]$. In any case, $\sigma_j' \geq 0$. For a training point $\mathbf{x}_i$, denote by $\sigma_{i,j}'$ the subgradient of $\left[\mathbf{w}_j^\top \mathbf{x}_i + b_j\right]_+$.

For all $j \in [k]$ that the partial derivatives of our 2-layer homogeneous neural network are given by

$$\frac{\partial}{\partial v_j} \Phi(\boldsymbol{\theta}; \mathbf{x}) = \left[\mathbf{w}_j^\top \mathbf{x} + b_j\right]_+,$$

$$\frac{\partial}{\partial \mathbf{w}_j} \Phi(\boldsymbol{\theta}; \mathbf{x}) = v_j \sigma_j' \mathbf{x},$$

$$\frac{\partial}{\partial b_j} \Phi(\boldsymbol{\theta}; \mathbf{x}) = v_j \sigma_j'.$$

Combining the above with the KKT conditions, we arrive at

$$v_j = \sum_{i=1}^n \lambda_i y_i \left[ \mathbf{w}_j^\top \mathbf{x}_i + b_j \right]_+ , \tag{7}$$

$$\mathbf{w}_j = v_j \sum_{i=1}^n \lambda_i y_i \mathbf{x}_i \sigma'_{i,j}, \tag{8}$$

$$b_j = v_j \sum_{i=1}^n \lambda_i y_i \sigma'_{i,j}, \tag{9}$$

for all $j \in [k]$.

## B  Proofs of lemmas and theorems in Section 3

We show an upper bound on the value $|\Phi(\boldsymbol{\theta}; \mathbf{x})|$ whenever $x$ is sampled according to $\mathcal{D}$ (that holds with high probability w.r.t. the initialization and $\mathbf{x}$) and a lower bound whenever $\mathbf{x}$ is in the training set (that holds with high probability w.r.t. the initialization). We prove that the lower bound is greater than the upper bound, thus giving us a way to differentiate between training and non training examples.

We use the same notations as in the previous section, which, for the sake of convenience, are specified again below.

**Notations**

- Let $J_+ = \{j : v_j > 0\}$ and $J_- = \{j : v_j < 0\}$.

- Let $m$ be the value of the network's margin (as defined in Assumption 2.1).

- Let $\delta = \max_{i \neq j} \left\{ |\mathbf{x}_i^\top \mathbf{x}_j| \right\}$, $\Delta = \min_{i \in [n]} \left\{ \|x_i\|^2 \right\}$ and $\Delta_{max} = \max_{i \in [n]} \{ \|x_i\|^2 \}$.

- For $\mathbf{x} \sim \mathcal{D}$ let $\delta_{\mathbf{x}} = \max\{\delta, \max_{i \in [n]} \{ |\mathbf{x}_i^\top \mathbf{x}| \}\}$.

**Lemma B.1.** *Under Assumption 3.1, with probability at least $1 - 2\tau$*

$$O\left(\frac{n \cdot \delta}{\Delta}\right) = o_d(1)$$

*Proof.* First, we prove, using the union bound, that $\Pr[n \cdot \delta \geq \Omega(d)] < \tau$. We have

$$\Pr[n \cdot \delta \geq \Omega(d)]$$

$$\leq \sum_{\substack{i,j=1 \\ i \neq j}}^n \Pr[n \cdot |\mathbf{x}_i^\top \mathbf{x}_j| \geq \Omega(d)] \leq \binom{n}{2} \cdot \frac{\tau}{n^2} < \tau$$

Secondly, we prove using the union bound that $\Pr[\Delta < o(d)] < \tau$.

$$\Pr[\Delta < o(d)] \leq \sum_{i=1}^n \Pr[\|\mathbf{x}_i\|^2 < o(d)] \leq n \cdot \frac{\tau}{n} = \tau.$$

Now, using the union bound again, we get

$$\Pr\left[\frac{n \cdot \delta}{\Delta} > \Omega_d(1)\right] \leq \Pr[\Delta < o(d)] + \Pr[n \cdot \delta \geq \Omega(d)] \leq 2\tau.$$

And hence with probability at least $1 - 2\tau$ we have that

$$O\left(\frac{n \cdot \delta}{\Delta}\right) = o_d(1)$$

$\square$

**Lemma B.2.** *Let $x \sim \mathcal{D}$. Under Assumption 3.1, with probability at least $1 - 2\tau$, we have*

$$O\left(\frac{n \cdot \delta_{\mathbf{x}}}{\Delta}\right) = o_d(1).$$

*Proof.* First, we prove, using the union bound, that $\Pr[\Delta < o(d)] < \tau$.

$$\Pr[\Delta < o(d)] \leq \sum_{i=1}^{n} \Pr[\|\mathbf{x}_i\|^2 < o(d)] \leq n \cdot \frac{\tau}{n} = \tau.$$

Second, we prove using another union bound that $\Pr[n \cdot \delta_{\mathbf{x}} \geq \Omega(d)] < \tau$. We have

$$\Pr[n \cdot \delta_{\mathbf{x}} \geq \Omega(d)]$$

$$\leq \sum_{\substack{i,j=1 \\ i \neq j}}^{n} \Pr[n \cdot |\mathbf{x}_i^\top \mathbf{x}_j| \geq \Omega(d)] + \sum_{i=1}^{n} \Pr[n \cdot |\mathbf{x}_i^\top \mathbf{x}| \geq \Omega(d)]$$

$$\leq \binom{n+1}{2} \cdot \frac{\tau}{n^2} < \tau.$$

Where in last inequality we used the fact that $n \geq 3$. Now, using the union bound again, we get

$$\Pr\left[\frac{n \cdot \delta_{\mathbf{x}}}{\Delta} > \Omega_d(1)\right] \leq \Pr[\Delta < o(d)] + \Pr[n \cdot \delta_{\mathbf{x}} \geq \Omega(d)] \leq 2\tau.$$

And hence with probability at least $1 - 2\tau$ we have that

$$O\left(\frac{n \cdot \delta_{\mathbf{x}}}{\Delta}\right) = o_d(1).$$

$\square$

The following 2 lemmas follow arguments from Frei et al. (2023b).

**Lemma B.3.** *With probability at least $1 - 2\tau$ over the choice of the training set, for all $l \in [n]$ we have*

$$\max\left\{\sum_{j \in J_+} v_j^2 \lambda_l \sigma_{l,j}', \sum_{j \in J_-} v_j^2 \lambda_l \sigma_{l,j}'\right\} \leq \frac{m}{\Delta + 1 - 2(\delta + 1)(n-1)}.$$

*Proof.* W.l.o.g. $\max_{i \in [n]} \left(\sum_{j \in J_+} v_j^2 \lambda_i \sigma_{i,j}'\right) \geq \max_{i \in [n]} \left(\sum_{j \in J_-} v_j^2 \lambda_i \sigma_{i,j}'\right)$ (the other direction is similar). Denote $\alpha = \max_{i \in [n]} \left(\sum_{j \in J_+} v_j^2 \lambda_i \sigma_{i,j}'\right)$ and $k \in \arg\max_{i \in [n]} \left(\sum_{j \in J_+} v_j^2 \lambda_i \sigma_{i,j}'\right)$.
If $\lambda_k = 0$, we are done. Otherwise, by KKT we know that $y_k \Phi(\boldsymbol{\theta}; x_k) = m$.
By Eq. (8) and Eq. (9) we have for all $j$

$$\mathbf{w}_j^\top \mathbf{x}_k + b_j = \sum_{i=1}^{n} \lambda_i y_i \sigma_{i,j}' v_j (\mathbf{x}_i^\top \mathbf{x}_k + 1)$$

$$= \lambda_k y_k \sigma_{k,j}' v_j (\|\mathbf{x}_k\|^2 + 1) + \sum_{i \neq k} \lambda_i y_i \sigma_{i,j}' v_j (\mathbf{x}_i^\top \mathbf{x}_k + 1) \qquad (10)$$

Consider 2 cases:
**CASE 1:** assume $y_k = 1$.

$$m = y_k \Phi(\boldsymbol{\theta}; \mathbf{x}_k)$$

$$= \sum_{i=1}^{n} v_i \left[\mathbf{w}_i^\top \mathbf{x}_k + b_i\right]_+$$

$$\geq \sum_{j \in J_+} v_j (\mathbf{w}_j^\top \mathbf{x}_k + b_j) + \sum_{j \in J_-} v_j \left[\mathbf{w}_j^\top \mathbf{x}_k + b_j\right]_+. \qquad (11)$$

Using the assumption that $y_k = 1$ and Eq. (10) we get

$$\sum_{j \in J_+} v_j(\mathbf{w}_j^\top \mathbf{x}_k + b_j) = \sum_{j \in J_+} \left( \lambda_k \sigma'_{k,j} v_j^2 (\|\mathbf{x}_k\|^2 + 1) + \sum_{i \neq k} \lambda_i y_i \sigma'_{i,j} v_j^2 (\mathbf{x}_i^\top \mathbf{x}_k + 1) \right)$$

$$\geq \sum_{j \in J_+} \lambda_k v_j^2 \sigma'_{k,j} (\Delta + 1) - (\delta + 1) \sum_{j \in J_+} \sum_{i \neq k} \lambda_i \sigma'_{i,j} v_j^2$$

$$\geq (\Delta + 1)\alpha - (\delta + 1)(n-1)\alpha. \tag{12}$$

Using $y_k = 1$ and Eq. (10) again we get

$$\sum_{j \in J_-} v_j \left[ \mathbf{w}_j^\top \mathbf{x}_k + b_j \right]_+ = \sum_{j \in J_-} v_j \left[ \lambda_k y_k \sigma'_{k,j} v_j (\|x_k\|^2 + 1) + \sum_{i \neq k} \lambda_i y_i \sigma'_{i,j} v_j (\mathbf{x}_i^\top \mathbf{x}_k + 1) \right]_+$$

$$\geq \sum_{j \in J_-} v_j \left[ \sum_{i \neq k} \lambda_i y_i \sigma'_{i,j} v_j (\mathbf{x}_i^\top \mathbf{x}_k + 1) \right]_+$$

$$\geq \sum_{j \in J_-} v_j \left[ \sum_{i \neq k} \lambda_i \sigma'_{i,j} |v_j| \cdot |\mathbf{x}_i^\top \mathbf{x}_k + 1| \right]_+$$

$$\geq \sum_{j \in j_-} v_j \left[ \sum_{i \neq k} \lambda_i \sigma'_{i,j} |v_j| (\delta + 1) \right]_+$$

$$= -(\delta + 1) \sum_{j \in j_-} \sum_{i \neq k} \lambda_i \sigma'_{i,j} v_j^2 \geq -(\delta + 1)(n-1)\alpha. \tag{13}$$

Combining Eq. (11), Eq. (12) and Eq. (13) we get

$$m \geq (\Delta + 1)\alpha - (\delta + 1)(n-1)\alpha - (\delta + 1)(n-1)\alpha$$

$$= \alpha(\Delta + 1 - 2(\delta + 1)(n-1))$$

$$\Rightarrow \alpha \leq \frac{m}{\Delta + 1 - 2(\delta + 1)(n-1)}.$$

**CASE 2:** Assume $y_k = -1$.
First, we show that for every $j \in J_+$ we have

$$\lambda_k \sigma'_{k,j} v_j \leq \frac{\delta + 1}{\Delta + 1} \sum_{i \neq k} \lambda_i \sigma'_{i,j} v_j.$$

Fix some $j \in J_+$. If $\sigma'_{j,k} = 0$ then

$$\lambda_k \sigma'_{k,j} v_j = 0 \leq \frac{\delta + 1}{\Delta + 1} \sum_{i \neq k} \lambda_i \sigma'_{i,j} v_j.$$

Otherwise, by the definition of $\sigma'_{k,j}$ we have $\mathbf{w}_j^\top \mathbf{x}_k + b_j \geq 0$.

$$0 \leq \mathbf{w}_j^\top \mathbf{x}_k + b_j = \sum_{i \neq k} \lambda_i y_i \sigma'_{i,j} v_j (\mathbf{x}_i^\top \mathbf{x}_k + 1) + \lambda_k y_k \sigma'_{k,j} v_j (\|\mathbf{x}_k\|^2 + 1)$$

$$\leq \sum_{i \neq k} \lambda_i \sigma'_{i,j} v_j (\delta + 1) - \lambda_k \sigma'_{k,j} v_j (\Delta + 1)$$

$$\Rightarrow \lambda_k \sigma'_{k,j} v_j \leq \frac{\delta + 1}{\Delta + 1} \sum_{i \neq k} \lambda_i \sigma'_{i,j} v_j.$$

Now we can upper bound the sum $\sum_{j \in J_+} \lambda_k \sigma'_{k,j} v_j^2$.

$$
\sum_{j \in J_+} \lambda_k \sigma'_{k,j} v_j^2 \leq \frac{\delta + 1}{\Delta + 1} \sum_{j \in J_+} \sum_{i \neq k} \lambda_i \sigma'_{i,j} v_j^2
$$

$$
\leq \frac{\delta + 1}{\Delta + 1} (n - 1) \cdot \max_{i \in [n]} \left( \sum_{j \in J_+} \lambda_i \sigma'_{i,j} v_j^2 \right)
$$

$$
< \max_{i \in [n]} \left( \sum_{j \in J_+} \lambda_i \sigma'_{i,j} v_j^2 \right) = \sum_{j \in J_+} \lambda_k \sigma'_{k,j} v_j^2
$$

Where the last inequality holds with probability at least $1 - 2\tau$ (using B.1 and the last equality in the definition of $k$). We get $\sum_{j \in J_+} \lambda_k \sigma'_{k,j} v_j^2 < \sum_{j \in J_+} \lambda_k \sigma'_{k,j} v_j^2$ - a contradiction. Consequently, $y_k = 1$ and we have already proved that case. $\square$

**Lemma B.4.** *With probability at least $1 - 2\tau$ over the choice of the training set, for all $l \in [n]$ such that $y_l = 1$ we have*

$$
\sum_{j \in J_+} v_j^2 \lambda_l \sigma'_{l,j} \geq \left( m - (\delta + 1)(n - 1) \frac{m}{\Delta + 1 - 2(\delta + 1)(n - 1)} \right) \cdot \frac{1}{\Delta_{max} + 1},
$$

*and for all $l \in [n]$ such that $y_l = -1$ we have*

$$
\sum_{j \in J_-} v_j^2 \lambda_l \sigma'_{l,j} \geq \left( m - (\delta + 1)(n - 1) \frac{m}{\Delta + 1 - 2(\delta + 1)(n - 1)} \right) \cdot \frac{1}{\Delta_{max} + 1}.
$$

*Proof.* We begin by showing the above for $J_+$ first. Let $k \in [n]$ be such that $y_k = 1$. We have

$$
m \leq \Phi(\boldsymbol{\theta}; \mathbf{x}_k)
$$

$$
= \sum_{j \in J} v_j \left[ \mathbf{w}_j^\top \mathbf{x}_k + b_j \right]_+
$$

$$
\leq \sum_{j \in J+} v_j \left[ \mathbf{w}_j^\top \mathbf{x}_k + b_j \right]_+
$$

$$
\leq \sum_{j \in J_+} v_j | \mathbf{w}_j^\top \mathbf{x}_k + b_j |.
$$

Let us bound it from above as follows

$$
\sum_{j \in J_+} v_j \left| \lambda_k y_k \sigma'_{k,j} v_j (\|\mathbf{x}_k\|^2 + 1) + \sum_{i \neq k} \lambda_i y_i \sigma'_{i,j} v_j (\mathbf{x}_i^\top \mathbf{x}_k + 1) \right| \leq \sum_{j \in J_+} v_j \left( \lambda_k \sigma'_{k,j} v_j (\|\mathbf{x}_k\|^2 + 1) + \sum_{i \neq k} \lambda_i \sigma'_{i,j} v_j |\mathbf{x}_i^\top \mathbf{x}_k + 1| \right)
$$

$$
= \sum_{j \in J_+} \left( \lambda_k \sigma'_{k,j} v_j^2 (\|\mathbf{x}_k\|^2 + 1) + \sum_{i \neq k} \lambda_i \sigma'_{i,j} v_j^2 |\mathbf{x}_i^\top \mathbf{x}_k + 1| \right)
$$

$$
\leq \sum_{j \in J_+} \left( (\Delta_{max} + 1) \lambda_k \sigma'_{k,j} v_j^2 + (\delta + 1) \sum_{i \neq k} \lambda_i \sigma'_{i,j} v_j^2 \right)
$$

$$
= (\Delta_{max} + 1) \sum_{j \in J_+} \lambda_k \sigma'_{k,j} v_j^2 + (\delta + 1) \sum_{j \in J_+} \sum_{i \neq k} \lambda_i \sigma'_{i,j} v_j^2
$$

Using B.3 we get with probability as least $1 - 2\tau$

$$m \leq (\Delta_{max} + 1) \sum_{j \in J_+} \lambda_k \sigma'_{k,j} v_j^2 + (\delta + 1)(n - 1)\frac{m}{\Delta + 1 - 2(\delta + 1)(n - 1)}$$

$$\Rightarrow \sum_{j \in J_+} \lambda_k \sigma'_{k,j} v_j^2 \geq \left( m - (\delta + 1)(n - 1)\frac{m}{\Delta + 1 - 2(\delta + 1)(n - 1)} \right) \cdot \frac{1}{\Delta_{max} + 1}.$$

Analogous arguments yield the same inequality for $J_-$. □

*Proof of Thm. 3.2.* Assume that $\mathbf{x}$ is in the training data, i.e. there exists $k \in [n]$ such that $\mathbf{x} = \mathbf{x}_k$. Assume w.l.o.g. that $y_k \Phi(\boldsymbol{\theta}; \mathbf{x}_k) > 0$, i.e. $y_k = 1$ (the case $y_k = -1$ is similar).

With probability of at least $1 - 2\tau$ Lemma B.1, Lemma B.3 and Lemma B.4 hold. From Lemma B.4 we have that

$$\sum_{j \in J_+} v_j^2 \lambda_k \sigma'_{k,j} \geq \left( m - (\delta + 1)(n - 1)\frac{m}{\Delta + 1 - 2(\delta + 1)(n - 1)} \right) \cdot \frac{1}{\Delta_{max} + 1}$$

By Lemma B.1 we have

$$O\left(\frac{n \cdot \delta}{\Delta}\right) = o_d(1),$$

which means that

$$\left( m - (\delta + 1)(n - 1)\frac{m}{\Delta + 1 - 2(\delta + 1)(n - 1)} \right) \cdot \frac{1}{\Delta_{max} + 1} > 0,$$

implying that $\lambda_k > 0$ (since otherwise the above sum will equal zero), and also that $\mathbf{x}_k$ is on the margin, and hence $\Phi(\boldsymbol{\theta}; \mathbf{x}_k) = m$.

If $\mathbf{x} \sim \mathcal{D}$, then

$$|\Phi(\boldsymbol{\theta}; x)|$$
$$= \left| \sum_{j \in J_+} v_j \sum_{i \in [n]} \lambda_i y_i \sigma'_{i,j} v_j (\mathbf{x}_i^\top \mathbf{x} + 1) + \sum_{j \in J_-} v_j \sum_{i \in [n]} \lambda_i y_i \sigma'_{i,j} v_j (\mathbf{x}_i^\top \mathbf{x} + 1) \right|$$
$$\leq \sum_{j \in J_+} \sum_{i \in [n]} \lambda_i \sigma'_{i,j} v_j^2 |\mathbf{x}_i^\top \mathbf{x} + 1| + \sum_{j \in J_-} \sum_{i \in [n]} \lambda_i \sigma'_{i,j} v_j^2 |\mathbf{x}_i^\top \mathbf{x} + 1|$$
$$\leq 2 \cdot n \cdot (\delta_\mathbf{x} + 1) \cdot \frac{m}{\Delta + 1 - 2(\delta + 1)(n - 1)} = O\left(\frac{n \cdot m \cdot \delta_\mathbf{x}}{\Delta}\right)$$

Where the second inequality holds by Lemma B.3 and last equality holds by Lemma B.1. By Lemma B.2 we have with probability at least $1 - 2\tau$

$$O\left(\frac{n \cdot m \cdot \delta_\mathbf{x}}{\Delta}\right) = m \cdot O\left(\frac{n \cdot \delta_\mathbf{x}}{\Delta}\right) = o_d(m)$$

Using the union bound on the previous events, we have that with probability at least $1 - 4\tau$, if $\mathbf{x} \sim \mathcal{D}$ then $|\Phi(\boldsymbol{\theta}; \mathbf{x})| = o_d(m)$. □

**Remark B.5** (On the lower bound of the margin)**.** *From Thm. 3.2 we know that w.h.p. at least $\frac{n}{2}$ training points lie on the margin. Our loss function is*

$$\ell(\Phi(\boldsymbol{\theta}; \mathbf{x}) \cdot y) = \log(1 + e^{-y \cdot \Phi(\boldsymbol{\theta}; \mathbf{x})})$$

*so we have that*

$$\frac{1}{2e} > L(\boldsymbol{\theta}) = \frac{1}{n} \sum_{i=1}^{n} \ell(\Phi(\mathbf{x}_i), y_i) \geq \frac{1}{n} \cdot \frac{n}{2} \cdot \log(1 + e^{-m})$$

$$\Rightarrow \frac{1}{e} > \log(1 + e^{-m})$$

*Now we can derive a lower bound on m:*

$$\log(1 + e^{-m}) < \frac{1}{e} \Rightarrow 1 + e^{-m} < e^{e^{-1}} \Rightarrow e^{-m} < e^{e^{-1}} \Rightarrow m > \frac{1}{e}.$$

*An analogous argument shows a similar bound for the exponential loss $\ell(x) = e^{-x}$.*

**Remark B.6** (On the upper bound of the margin)**.** *When training a neural network using gradient-based methods, the training process usually halts once the gradient is sufficiently small. When considering the exponential or logistic losses as in our case, a large margin implies a small loss, which in turn implies that the gradient is small. This suggests that making further progress when the margin is large becomes very difficult, and the training process is expected to stop. More formally, recall the logistic loss function (a similar argument implies the same result for the exponential loss):*

$$\ell(\Phi(\boldsymbol{\theta}; \mathbf{x}) \cdot y) = \log(1 + e^{-y \cdot \Phi(\boldsymbol{\theta}; \mathbf{x})}).$$

*This function is monotonically decreasing in the expression $y\Phi(\boldsymbol{\theta}; \mathbf{x})$, so the loss is maximized for points that are on the margin, and we can upper bound*

$$\left| \frac{\partial \ell(\Phi(\boldsymbol{\theta}; \mathbf{x}) \cdot y)}{\partial \Phi(\boldsymbol{\theta}; \mathbf{x})} \right| = \left| \frac{-y \cdot \Phi(\boldsymbol{\theta}; \mathbf{x}) \cdot e^{-y \cdot \Phi(\boldsymbol{\theta}; \mathbf{x})}}{1 + e^{-y \cdot \Phi(\boldsymbol{\theta}; \mathbf{x})}} \right| \leq \left| \frac{me^{-m}}{1 + e^{-m}} \right|.$$

*The above yields*

$$\left| \frac{\partial L(\boldsymbol{\theta})}{\partial \boldsymbol{\theta}_j} \right| \leq \frac{1}{n} \sum_{i=1}^{n} \left| \frac{\partial \ell(\Phi(\boldsymbol{\theta}; \mathbf{x}_i) \cdot y_i)}{\partial \Phi(\boldsymbol{\theta}; \mathbf{x}_i)} \right| \cdot \left| \frac{\partial \Phi(\boldsymbol{\theta}; \mathbf{x}_i)}{\partial \boldsymbol{\theta}_j} \right| \leq \mathrm{poly}(d) \cdot \left| \frac{me^{-m}}{1 + e^{-m}} \right|,$$

*which allows us to bound the norm of the gradient by:*

$$\|\nabla_{\boldsymbol{\theta}} L(\boldsymbol{\theta})\| \leq w \cdot \mathrm{poly}(d) \cdot \left| \frac{me^{-m}}{1 + e^{-m}} \right| = \mathrm{poly}(d) \cdot \left| \frac{me^{-m}}{1 + e^{-m}} \right|,$$

*where $w$ denotes the width of the network which we assume to be polynomial in $d$ (since otherwise even making a prediction is computationally inefficient).*

*If, for example, the margin is $m = \log^2 d = o(\sqrt{d})$, we get that*

$$\|\nabla_{\boldsymbol{\theta}} L(\boldsymbol{\theta})\| \leq \mathrm{poly}(d) \left| \frac{\log^2 d e^{-\log^2 d}}{1 + e^{-\log^2 d}} \right| \leq \mathrm{poly}(d) \log^2 d \cdot e^{-\log^2 d} = \mathrm{poly}(d) \log^2 d \cdot d^{-\log d},$$

*which is smaller than any inverse polynomial in $d$. Hence, if we train for at most polynomially many iterations and label all the data points correctly (i.e. the margin is strictly positive), then training effectively stops when the margin reaches $O(\log^2 d) = o(\sqrt{d})$, and all the data points on the margin (which consist of at least one point) will have an output of magnitude $O(polylog(d))$.*

## C  High-dimensional attacks in the statistically learnable case

In this appendix, we show that Item 3 exemplifies a setting where Assumption 3.1 is satisfied, yet the distribution being considered is statistically learnable. This was shown in several recent works, which considered the optimization of a shallow neural network, in a setting similar to ours.

Consider, for example, the setting studied in Xu et al. (2023). In that paper, the authors prove a generalization result under the assumption of a certain target distribution of a mixture of four Gaussians. Such a distribution is captured by Item 3 in our examples for distributions which satisfy Assumption 3.1, which indicates that our proposed membership inference attack will work. Specifically, to make sure that both Assumption 3.1 and the requirements made in Xu et al. (2023) are satisfied, it must also hold that:

- The norm of each mean satisfies $\|\boldsymbol{\mu}^{(i)}\|^2 \geq \Omega(n^{0.51}\sqrt{d})$.

- The dimension of the feature space satisfies $d \geq \Omega(n^2 \max\{\|\boldsymbol{\mu}^{(i)}\|^2\})$.

- The number of neurons satisfies $k \geq \Omega(n^{0.02})$.

Quite more precisely, their theorem states the following:

**Theorem C.1** (Xu et al. (2023), Theorem 3.1, informal). *Suppose that the above assumptions are satisfied, then with high probability over the training set and the initialization of the weights, we have*

$$\Pr_{(\mathbf{x},y)\sim\mathcal{D}}[y \neq \text{sign}(\Phi(\boldsymbol{\theta};\mathbf{x}))] \leq \exp(-\Omega(n^{2.01}))$$

These assumptions essentially imply Assumption 3.1.

Similarly, Assumption 3.1, and specifically Item 3 in our examples, also holds in other settings where generalization was proved in previous works:

- Xu and Gu (2023); Frei et al. (2022); Chatterji and Long (2021) proved generalization in a setting where the data distribution consists of two opposite Gaussians (or more broadly in an even more general setting) with covariance $I_d$ and means $\pm\boldsymbol{\mu}$, where $\|\boldsymbol{\mu}\| = d^\beta$ with $\beta \in (0.25, 0.5)$. Their sample size is $n = \tilde{\Omega}(1)$. This setting satisfies our condition from Item 3. Specifically, the result of Xu and Gu (2023) holds for 2-layer ReLU networks.

- In Frei et al. (2023a) (see the discussion after Theorem 11 therein), the authors mention two specific settings that satisfy their theorem requirements, and thus good generalization performance can be achieved (and more specifically, in Corollaries 12 and 13, they further show that in these settings good generalization is achieved by the max-margin linear predictor and by a trained 2-layer leaky-ReLU network). Note that these settings satisfy our condition from Item 3.

## D  Proofs of distributions

In this section we prove the examples in Section 3.

**Uniform Distribution**  For the uniform distribution on $\sqrt{d} \cdot \mathbb{S}^{d-1}$, the next lemma shows why is satisfies our assumptions.
The lemma is from Vardi et al. (2022a), and we give a paraphrased version of it for the reader's convenience.

**Lemma D.1.** *Let $\mathbf{x}, \mathbf{y} \sim U(\sqrt{d} \cdot \mathbb{S}^{d-1})$. Then, with probability at least $1 - d^{1-\ln(d)/4} = 1 - o_d(1)$ we have $|\langle \mathbf{x}, \mathbf{y} \rangle| \leq \sqrt{d} \cdot \log d = o(d)$.*

**Remark D.2.** *For the uniform distribution, the training set size can be $n = o\left(\frac{\sqrt{d}}{\log d}\right)$ and we have $\tau = n^2 \cdot d^{1-\ln(d)/4} = o_d(1)$*

**Normal Distribution**  As for the normal distribution, the following two lemmas prove its correctness.

**Lemma D.3.** *Let $\mathcal{N} = \mathcal{N}(\boldsymbol{\mu}, I)$ be a normal distribution on $\mathbb{R}^d$. Let $\mathbf{x}, \mathbf{y} \sim \mathcal{N}(\mu, I)$. Assume that $\|\boldsymbol{\mu}\|^2 = o(d)$. then with probability at least*

$$1 - 2\exp\left(-\frac{c_1}{16c_2^2} \cdot \frac{d^{2\epsilon}}{\|\mu\|^2}\right) - \max\left(2\exp\left(-\frac{c_1}{2c_2^2}d^\epsilon\right), 2\exp\left(-\frac{c_1}{4c_2^4} \cdot d^{2\epsilon-1}\right)\right)$$

$$- \max\left(2\exp\left(-\frac{c_1}{c_2^4}d^{2\epsilon-1}\right), 2\exp\left(-\frac{c_1}{c_2^2}d^\epsilon\right)\right)$$

$$= 1 - o_d(1)$$

*we have $|\langle \mathbf{x}, \mathbf{y} \rangle| = o(d)$ and $\|\mathbf{x}\|^2 = O(d)$, where $c_1$, $c_2$ are constants independent of $d$, and $\frac{1}{2} < \epsilon < 1$.*

*Proof.* Let $\mathbf{x}, \mathbf{y} \sim \mathcal{N}(\boldsymbol{\mu}, \Sigma)$ independently.
w.l.o.g. $\Sigma$ is diagonal, otherwise there is a unitary matrix $U$ such that $U\mathbf{x}, U\mathbf{y} \sim \mathcal{N}(U\boldsymbol{\mu}, U\Sigma U^\top)$ where $U\Sigma U^\top$ is diagonal. Since $U$ is unitary we have that

$$\langle U\mathbf{x}, U\mathbf{y} \rangle = \langle \mathbf{x}, \mathbf{y} \rangle$$
$$\|U\mathbf{x}\| = \|\mathbf{x}\|$$

So we can assume that $\Sigma$ is diagonal.
For comfort, we define some notations:

- The sub-Gaussian norm $\|\cdot\|_{\psi_2}$ for a sub-Gaussian random variable $\mathbf{x}$ is defined by

$$\|\mathbf{x}\|_{\psi_2} = \inf\left\{t > 0 : E\left[\exp\left(\frac{\mathbf{x}^2}{t}\right)\right] \leq 2\right\}$$

- The sub-exponential norm $\|\cdot\|_{\psi_1}$ for a sub-exponential random variable $\mathbf{x}$ is defined by

$$\|\mathbf{x}\|_{\psi_1} = \inf\left\{t > 0 : E\left[\exp\left(\frac{|\mathbf{x}|}{t}\right)\right] \leq 2\right\}$$

First, let us compute $E\left[\|\mathbf{x}\|^2\right]$. Note that

$$\|\mathbf{x}\|^2 = \sum_{i=1}^d \mathbf{x}_i^2,$$

then $E[\mathbf{x}_i^2] = E[\mathbf{x}_i]^2 + \text{Var}(\mathbf{x}_i) = \boldsymbol{\mu}_i^2 + 1$

$$E\left[\|\mathbf{x}\|^2\right] = E\left[\sum_{i=1}^d \mathbf{x}_i^2\right] = \sum_{i=1}^d E[\mathbf{x}_i^2] = \sum_{i=1}^d \text{Var}(\mathbf{x}_i) + \boldsymbol{\mu}_i^2 = \text{tr}(I) + \|\boldsymbol{\mu}\|^2 = O(d)$$

Note that we can write $\mathbf{x}$ as $\mathbf{x} = \boldsymbol{\mu} + \mathbf{z}$ where $\mathbf{z} \sim \mathcal{N}(0, I)$. We can write $\|\mathbf{x}\|^2 = \|\boldsymbol{\mu} + \mathbf{z}\|^2 = \|\boldsymbol{\mu}\|^2 + 2|\boldsymbol{\mu}^\top \mathbf{z}| + \|\mathbf{z}\|^2$. So we need to upper bound

$$\|\mathbf{x}\|^2 - E\left[\|\mathbf{x}\|^2\right] = \|\boldsymbol{\mu}\|^2 + 2\boldsymbol{\mu}^\top \mathbf{z} + \|\mathbf{z}\|^2 - \|\boldsymbol{\mu}\|^2 - 2\boldsymbol{\mu}^\top E[\mathbf{z}] - E\left[\|\mathbf{z}\|^2\right] = \|\mathbf{z}\|^2 - E\left[\|z\|^2\right] + 2\boldsymbol{\mu}^\top \mathbf{z}$$

Where in the last equality we used the fact that $E[\mathbf{z}] = 0$

From the union bound we get that for every $t > 0$

$$\Pr\left[\left|\mathbf{x}^2 - E[\|\mathbf{x}\|^2]\right| > t\right] = \Pr\left[\left|\|\mathbf{z}\|^2 - E[\|\mathbf{z}\|^2] + 2\boldsymbol{\mu}^\top \mathbf{z}\right| > t\right]$$

$$\leq \Pr\left[\left|\|\mathbf{z}\|^2 - E[\|\mathbf{z}\|^2]\right| + 2\left|\boldsymbol{\mu}^\top \mathbf{z}\right| > t\right]$$

$$\leq \Pr\left[\left|\|\mathbf{z}\|^2 - E[\|\mathbf{z}\|^2]\right| > \frac{t}{2}\right] + \Pr\left[2\left|\boldsymbol{\mu}^\top \mathbf{z}\right| > \frac{t}{2}\right]$$

Let us bound the first term. To do so, we use Hanson-Wright Inequality (Vershynin (2018) Theorem 6.2.1).

$$\Pr\left[\left|\|\mathbf{z}\|^2 - E[\|\mathbf{z}\|^2]\right| > \frac{t}{2}\right] \leq 2\exp\left[-c_1 \min\left(\frac{t^2}{4 \cdot K^4 \cdot d}, \frac{t}{2 \cdot K^2}\right)\right]$$

Where $K = \max_i \|\mathbf{x}_i\|_{\psi_2} = c_2$ and $c_1$, $c_2$ are constant independent of $d$. We set $t = d^\epsilon$ for $\frac{1}{2} < \epsilon < 1$.

**Case 1 - $\frac{t^2}{4 \cdot K^4 \cdot d}$ is the minimum**

$$\Pr\left[\left|\|\mathbf{z}\|^2 - E[\|\mathbf{z}\|^2]\right| > \frac{t}{2}\right] \le 2\exp\left(-c_1 \frac{d^{2\epsilon}}{c_2^4 \cdot 4 \cdot d}\right) = 2\exp\left(-\frac{c_1}{4 \cdot c_2^4} \cdot d^{2\epsilon-1}\right) = o_d(1)$$

**Case 2 - $\frac{t}{2 \cdot K^2}$ is the minimum**

$$\Pr\left[\left|\|\mathbf{z}\|^2 - E[\|\mathbf{z}\|^2]\right| > \frac{t}{2}\right] \le 2\exp\left(-c_1 \frac{d^{\epsilon}}{2 \cdot c_2^2}\right) = o_d(1)$$

Now we upper bound the term $\Pr\left[2|\boldsymbol{\mu}^\top \mathbf{z}| > \frac{t}{2}\right] = \Pr\left[|\boldsymbol{\mu}^\top \mathbf{z}| > \frac{t}{4}\right]$.
From General Hoeffding's inequality (Vershynin (2018) Theorem 2.6.3) we get that

$$\Pr\left[|\boldsymbol{\mu}^\top \mathbf{z}| > \frac{t}{4}\right] \le 2\exp\left(-\frac{c_1 t^2}{16 \cdot K^2 \cdot \|\boldsymbol{\mu}\|^2}\right)$$

Where $K = \max_i \|\mathbf{x}_i\|_{\psi_2} = c_2$ and $c_1$, $c_2$ are constant independent of $d$. Putting it all together we get

$$\Pr\left[|\boldsymbol{\mu}^\top \mathbf{z}| > \frac{t}{4}\right] \le 2\exp\left(-\frac{c_1 t^2}{16 \cdot K^2 \cdot \|\boldsymbol{\mu}\|^2}\right)$$
$$= 2\exp\left(-\frac{c_1}{16c_2^2}\frac{d^{2\epsilon}}{\|\boldsymbol{\mu}\|^2}\right) = 2\exp\left(-\frac{c_1}{16c_2^2}\frac{d^{2\epsilon}}{\|\boldsymbol{\mu}\|^2}\right) = o_d(1)$$

Where in last inequality we used the fact that $2\epsilon > 1$.

All in all, we showed that $E[\|\mathbf{x}\|^2] = O(d)$ and that with probability

$$1 - \max\left(2\exp\left(-\frac{c_1}{4c_2^2} \cdot d^{2\epsilon-1}\right), 2\exp\left(-\frac{c_1}{2c_2^2} \cdot d^{\epsilon}\right)\right) - 2\exp\left(-\frac{c_1}{16c_2^2}\frac{d^{2\epsilon}}{\|\mu\|^2}\right) = 1 - o_d(1)$$

we have that

$$\left|\|\mathbf{x}\|^2 - E[\|\mathbf{x}\|^2]\right| < d^{\epsilon} = o(d)$$

and specifically $\|\mathbf{x}\|^2 = O(d)$

Since $\mathbf{x}$ is normal, each $\mathbf{x}_i$ is sub-Gaussian (and the same for $\mathbf{y}$).
Let us have a look at $\mathbf{x}^\top \mathbf{y}$: Since $\mathbf{x}_i, \mathbf{y}_i$ are sub-Gaussians, $\mathbf{x}_i \cdot \mathbf{y}_i$ is sub-exponential (Vershynin (2018), Lemma 2.7.7). It is also known that a sum of sub-exponential random variables is in itself sub-exponential, so we get that

$$\mathbf{x}^\top \mathbf{y} = \sum_{i=1}^{d} x_i y_i$$

is sub-exponential. By the centering lemma (Vershynin (2018) Exercise 2.7.10), $x_i y_i - E[x_i y_i] = x_i y_i - \mu_i^2$ is also sub-exponential, with mean zero. We can use Bernstein's inequality (Vershynin (2018), Theorem 2.8.1) to get:

$$\Pr\left[|\mathbf{x}^\top \mathbf{y} - \|\boldsymbol{\mu}\|^2| > t\right] = \Pr\left[\left|\sum_{i=1}^{d} x_i y_i - \mu_i^2\right| > t\right]$$

$$\le 2\exp\left[-c_1 \cdot \min\left(\frac{t}{\max_i \|x_i y_i - \mu_i\|_{\psi_1}}, \frac{t^2}{\sum_{i=1}^{d} \|x_i y_i - \mu_i\|_{\psi_1}^2}\right)\right]$$

$$\le 2\exp\left[-c_1 \cdot \min\left(\frac{t}{\max_i \|x_i y_i\|_{\psi_1}}, \frac{t^2}{\sum_{i=1}^{d} \|x_i y_i\|_{\psi_1}^2}\right)\right]$$

$$\le 2\exp\left[-c_1 \cdot \min\left(\frac{t}{\max_i \|x_i\|_{\psi_2}\|y_i\|_{\psi_2}}, \frac{t^2}{\sum_{i=1}^{d} \|x_i\|_{\psi_2}^2\|y_i\|_{\psi_2}^2}\right)\right]$$

$$= 2\exp\left[-c_1 \cdot \min\left(\frac{t}{c_2^2}, \frac{t^2}{\sum_{i=1}^{d} c_2^4}\right)\right]$$

Where $c_1$, $c_2$ are constants that do not depend on the dimension $d$. In the second inequality we used the fact that $\|\mathbf{x} - E[\mathbf{x}]\|_{\psi_1} \leq \|\mathbf{x}\|_{\psi_1}$ (Vershynin (2018) Exercise 2.7.10) and in the third inequality we used the fact that $\|x_i y_i\|_{\psi_1} \leq \|x_i\|_{\psi_2} \|y_i\|_{\psi_2}$ (Vershynin (2018) Lemma 2.7.7). Setting $t = d^\epsilon$ for some $\frac{1}{2} < \epsilon < 1$ we get:

**Case 1 - $\frac{t}{c_2^2}$ is the minimum**

$$\Pr[|\mathbf{x}^\top \mathbf{y} - \|\boldsymbol{\mu}\|^2| > d^\epsilon] \leq 2\exp\left[-c_1 \cdot \frac{d^\epsilon}{c_2^2}\right] = o_d(1)$$

And since both $\|\boldsymbol{\mu}\|^2 = o(d)$ and $d^\epsilon = o(d)$ we get that w.h.p. $\mathbf{x}^\top \mathbf{y} = o(d)$

**Case 2 - $\frac{t^2}{\sum_{i=1}^d c_2^4}$ is the minimum**

$$\Pr[|\mathbf{x}^\top \mathbf{y} - \|\boldsymbol{\mu}\|^2| > d^\epsilon] \leq 2\exp\left[-c_1 \cdot \frac{d^{2\epsilon}}{c_2^4 \cdot d}\right]$$

$$= 2\exp\left[-\frac{c_1}{c_2^4} \cdot d^{2\epsilon-1}\right] = o_d(1)$$

Using the union bound, with probability at least

$$1 - 2\exp\left(-\frac{c_1}{16c_2^2} \cdot \frac{d^{2\epsilon}}{\|\mu\|^2}\right) - \max\left(2\exp\left(-\frac{c_1}{2c_2^2}d^\epsilon\right), 2\exp\left(-\frac{c_1}{4c_2^4} \cdot d^{2\epsilon-1}\right)\right)$$

$$- \max\left(2\exp\left(-\frac{c_1}{c_2^4}d^{2\epsilon-1}\right), 2\exp\left(-\frac{c_1}{c_2^2}d^\epsilon\right)\right)$$

$$= 1 - o_d(1)$$

we have $|\langle \mathbf{x}, \mathbf{y}\rangle| = o(d)$ and $\|\mathbf{x}\|^2 = O(d)$. $\qquad\square$

**Remark D.4.** *we want $n \cdot |\mathbf{x}^\top \mathbf{y}| = o(d)$ to hold, so*

$$n \cdot |\mathbf{x}^\top \mathbf{y}| \leq n \cdot (\|\mu\|^2 + d^\epsilon) = o(d) \Rightarrow n = \frac{o(d)}{\|\mu\|^2 + d^\epsilon}$$

**Lemma D.5.** *Let $\mathcal{N} = \mathcal{N}(\boldsymbol{\mu}, I)$ be a normal distribution on $\mathbb{R}^d$. Let $\mathbf{x}$, $\mathbf{y} \sim \mathcal{N}(\boldsymbol{\mu}, I)$. Assume that $\|\boldsymbol{\mu}\|^2 = o(d)$, and $n = \frac{o(d)}{\|\boldsymbol{\mu}\|^2 + d^\epsilon}$ for $\frac{1}{2} < \epsilon < 1$. Denote*

$$k = 2\exp\left(-\frac{c_1}{16c_2^2} \cdot \frac{d^{2\epsilon}}{\|\mu\|^2}\right) + \max\left(2\exp\left(-\frac{c_1}{2c_2^2}d^\epsilon\right), 2\exp\left(-\frac{c_1}{4c_2^4} \cdot d^{2\epsilon-1}\right)\right)$$

$$+ \max\left(2\exp\left(-\frac{c_1}{c_2^4}d^{2\epsilon-1}\right), 2\exp\left(-\frac{c_1}{c_2^2}d^\epsilon\right)\right)$$

*where $c_1$, $c_2$ are the constants from Lemma D.3. Let $\tau = k \cdot n$. Then with probability at least $1 - \frac{\tau}{n^2}$ have $|n \cdot \langle \mathbf{x}, \mathbf{y}\rangle| = o(d)$ and $\|\mathbf{x}\|^2 = O(d)$. In particular, those $n$ and $\tau$ satisfy Assumption 3.1.*

*Proof.* From Lemma D.3 we know that with probability at least $1 - k$ we have that $|\langle \mathbf{x}, \mathbf{y}\rangle| \leq \|\mu\|^2 + d^\epsilon$, so with probability at least $1 - k$ we have that $n \cdot |\langle \mathbf{x}, \mathbf{y}\rangle| = \frac{o(d)}{\|\mu\|^2 + d^\epsilon} \cdot |\langle \mathbf{x}, \mathbf{y}\rangle| \leq o(d)$. We also know from Lemma D.3 that with probability at least $1 - k$ we have that $\|\mathbf{x}\|^2 = \Omega(d)$. Setting $\tau = k \cdot n^2 = o_d(1)$ completes the proof. $\qquad\square$

**Mixture of $k$ Gaussians** We prove the case where we have 2 Gaussians, but the proof is similar for any number of Gaussians.

**Lemma D.6.** *Let $\mathcal{N} = \pi\mathcal{N}(\boldsymbol{\mu}^{(1)}, I) + (1-\pi)\mathcal{N}(\boldsymbol{\mu}^{(2)}, I)$ where $0 \leq \pi \leq 1$ be a mixture of normal distributions on $\mathbb{R}^d$. Assume the following:*

- $\|\boldsymbol{\mu}^{(1)}\|^2 = o(d)$, $\|\boldsymbol{\mu}^{(2)}\|^2 = o(d)$

- $n = \frac{o(d)}{\max(\|\mu^{(1)}\|^2, \|\mu^{(2)}\|^2) + d^\epsilon}$ *for $\frac{1}{2} < \epsilon < 1$.*

- $k$ *defined as in Lemma D.5*

- $\tau = k \cdot n^2$

*then with probability at least $1 - \frac{\tau}{n^2}$ we have $n \cdot |\langle \mathbf{x}, \mathbf{y} \rangle| = o(d)$ and $\|\mathbf{x}\|^2 = O(d)$*

*Proof.* Let $\mathbf{x}, \mathbf{y} \sim \pi\mathcal{N}(\boldsymbol{\mu}^{(1)}, I) + (1-\pi)\mathcal{N}(\boldsymbol{\mu}^{(2)}, I)$ where $0 \leq \pi \leq 1$. Let us compute $E\left[\|\mathbf{x}\|^2\right]$. We can think of $\mathbf{x}$ as

$$\mathbf{x} = \begin{cases} \mathbf{x}_1, & \text{with probability } \pi \\ \mathbf{x}_2, & \text{with probability } 1 - \pi \end{cases}$$

where $\mathbf{x}_1 \sim \mathcal{N}(\boldsymbol{\mu}^{(1)}, I)$ and $\mathbf{x}_2 \sim \mathcal{N}(\boldsymbol{\mu}^{(2)}, I)$. From the law of total expectation we get

$$E[\|\mathbf{x}\|^2] = \pi E[\|\mathbf{x}_1\|^2] + (1 - \pi)E[\|\mathbf{x}_2\|^2]$$

and from D.5 we get

$$E[\|\mathbf{x}\|^2] = \pi \cdot \left(\|\boldsymbol{\mu}^{(1)}\|^2 + \mathrm{tr}(I)\right) + (1 - \pi) \cdot \left(\|\boldsymbol{\mu}^{(2)}\|^2 + \mathrm{tr}(I)\right) = O(d)$$

Denote $A = \{\mathbf{x} : \left|\|\mathbf{x}\|^2 - E[\|\mathbf{x}\|]^2\right| > d^\epsilon\}$ where $\frac{1}{2} < \epsilon < 1$.
From the law of total probability we get:

$$p(A) = p(A|\mathbf{x} = \mathbf{x}_1) \cdot \pi + p(A|\mathbf{x} = \mathbf{x}_2) \cdot (1 - \pi)$$

$$= 1 - \max\left(2\exp\left(-\frac{c_1}{4c_2^2} \cdot d^{2\epsilon-1}\right), 2\exp\left(-\frac{c_1}{2c_2^2} \cdot d^\epsilon\right)\right) - 2\exp\left(-\frac{c_1}{16c_2^2}\frac{d^{2\epsilon}}{\|\mu\|^2}\right) = 1 - o_d(1)$$

and specifically, $\|\mathbf{x}\|^2 = O(d)$.

Now, let us show that $E[\mathbf{x}^\top \mathbf{y}] = o(d)$:

$$E[\mathbf{x}^\top \mathbf{y}] = E[\mathbf{x}^\top]E[\mathbf{y}] = \left(\pi\boldsymbol{\mu}^{(1)} + (1-\pi)\boldsymbol{\mu}^{(2)}\right)^\top \left(\pi\boldsymbol{\mu}^{(1)} + (1-\pi)\boldsymbol{\mu}^{(2)}\right)$$

$$= \pi^2\|\boldsymbol{\mu}^{(1)}\|^2 + 2\pi(1-\pi)\boldsymbol{\mu}^{(1)\top}\boldsymbol{\mu}^{(2)} + (1-\pi)^2\|\boldsymbol{\mu}^{(2)}\|^2$$

$$= \pi^2 o(d) + 2\pi(1-\pi)o(d) + (1-\pi)^2 o(d) = o(d)$$

We divide the proof into 4 cases.

**Case 1**: $\mathbf{x}, \mathbf{y} \sim \mathcal{N}(\boldsymbol{\mu}^{(1)}, I)$

In this case, both points came from the same normal distribution, which we have already proven.

**Case 2**: $\mathbf{x} \sim \mathcal{N}(\boldsymbol{\mu}^{(1)}, I)$ and $\mathbf{y} \sim \mathcal{N}(\boldsymbol{\mu}^{(2)}, I)$

For every $i$ we have that $x_i$ and $y_i$ are sub-Gaussians and $\|x_i\|_{\psi_2} \leq c$, $\|y_i\|_{\psi_2} \leq c$, so we can use the same logic as in Lemma D.3 do prove that $\mathbf{x}^\top \mathbf{y} = o(d)$ with the same probability.

**Case 3**: $\mathbf{x} \sim \mathcal{N}(\boldsymbol{\mu}^{(2)}, I)$ and $\mathbf{y} \sim \mathcal{N}(\boldsymbol{\mu}^{(1)}, I)$

Same as case 2.

**Case 4**: $\mathbf{x} \sim \mathcal{N}(\boldsymbol{\mu}^{(2)}, I)$ and $\mathbf{y} \sim \mathcal{N}(\boldsymbol{\mu}^{(2)}, I)$

Same as case 1

Similar to D.4, with probability at least $1 - k$ we have that

$$n \cdot \langle \mathbf{x}, \mathbf{y} \rangle \leq n \cdot \max(\|\mu^{(1)}\|^2, \|\mu^{(2)}\|^2) + d^\epsilon = o(d) \Rightarrow n = \frac{o(d)}{\max\left\{\|\boldsymbol{\mu}^{(1)}\|^2, \|\boldsymbol{\mu}^{(2)}\|^2\right\} + d^\epsilon}$$

and also that $\|\mathbf{x}\|^2 = \Omega(d)$. Setting $\tau = k \cdot n^2 = o_d(1)$ completes the proof. $\qquad\square$

## E  Experiments For MIA For High Dimensional Data

In this section, we empirically demonstrate the membership inference attack in settings much broader than those specified in Assumption 3.1. In all experiments except those using the MNIST dataset, the training set consisted of 20 samples, and the test set consisted of $5,000$ samples. All samples were sampled using the Gaussian distribution, following the experiment in 3.2. All experiments achieved 100% training accuracy and over 85% test accuracy.

### E.1  MNIST

We conducted experiments on the MNIST dataset. The first experiment (Figure 2) employed the same fully connected architecture described in Section 3.2, while the second (Figure 3) used a convolutional neural network consisting of two convolutional layers with 32 and 64 channels, respectively, each followed by ReLU activations, and a final fully connected layer. To vary the input dimension, we resized the images using bicubic interpolation. We trained the network for 4000 epochs.

The empirical results are consistent with our theoretical predictions. Specifically, as the input dimension increases, the proportion of training samples that lie on the margin increases, whereas the proportion of test samples whose output value is greater than or equal to the margin decreases. This behavior is observed for both the fully connected and convolutional architectures.

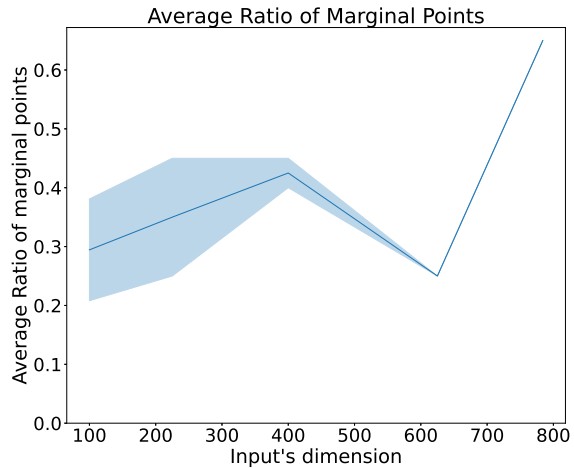
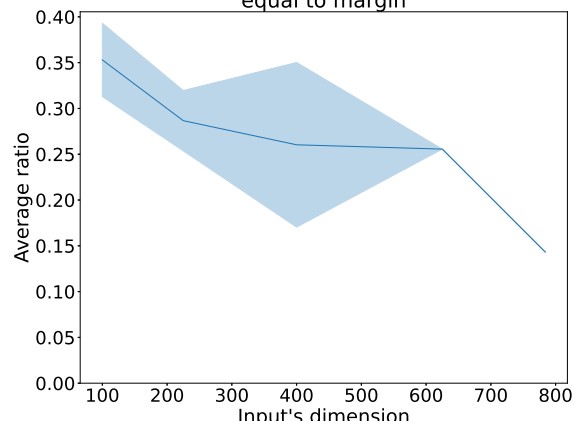

(a) The percentage of training points that lie on the margin (up to 10% slack) increases with the dimension.

(b) The percentage of test points that lie on or above the margin decreases as the dimension increases.

Figure 2: Results for the fully connected architecture on MNIST as a function of the input dimension. As the dimension increases, a larger fraction of training points lie on the margin (a), while a smaller fraction of test points lie on or above the margin (b). Results are averaged over 10 independent runs.

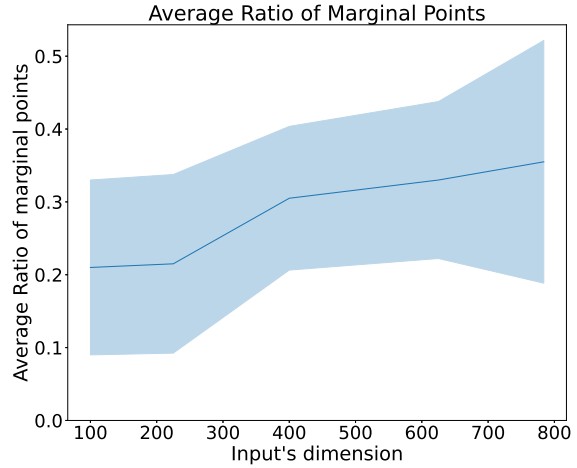 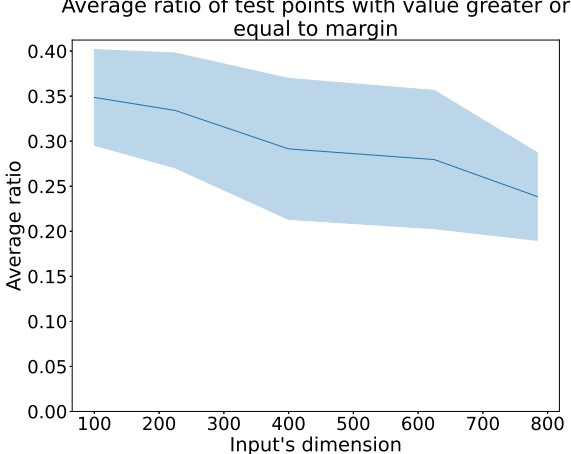

(a) The percentage of training points that lie on the margin (up to 10% slack) increases with the dimension.

(b) The percentage of test points that lie on or above the margin decreases as the dimension increases.

Figure 3: Results for the convolutional architecture on MNIST as a function of the input dimension. As the dimension increases, a larger fraction of training points lie on the margin (a), while a smaller fraction of test points lie on or above the margin (b). Results are averaged over 10 independent runs.

## E.2   Approximate KKT

Our theoretical analysis assumes that the network has converged to a KKT point. In this experiment, we investigate the effectiveness of the proposed MIA as a function of the network's proximity to a KKT point. We use the same architecture and Gaussian data distribution described in Section 3.2. The data dimension is 500 and the training set consists of 20 samples. During training, we evaluate after each epoch the proportion of marginal points in both the training and test sets. The x-axis in Figure 4 corresponds to the number of training epochs, which serves as a proxy for the network's proximity to a KKT point.

The results indicate that the attack becomes increasingly effective as training progresses and the network approaches a KKT point. Specifically, the proportion of training samples lying on the margin increases, while the proportion of test samples whose output value exceeds the margin decreases. Nevertheless, the attack remains effective even before full convergence is reached. In particular, the false positive rate decreases rapidly and approaches zero after a relatively small number of training epochs (Figure 4b).

## E.3   Network's width

Our theoretical analysis suggests that the effectiveness of the proposed membership inference attack is independent of the network width. To empirically verify it, we trained two-layer fully connected neural networks with varying hidden-layer widths. The results are presented in Figure 5.

Consistent with our theory, the proportion of training samples that lie on the margin remains essentially unchanged as the network width increases. Similarly, the proportion of test samples whose output value lies on or above the margin remains negligible and exhibits no systematic dependence on the width. These findings suggest that the effectiveness of the attack is essentially independent of network width, in agreement with our theoretical results.

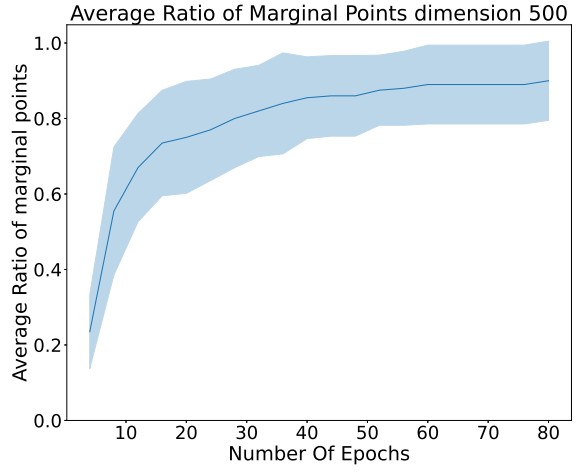

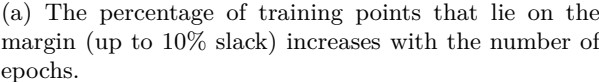

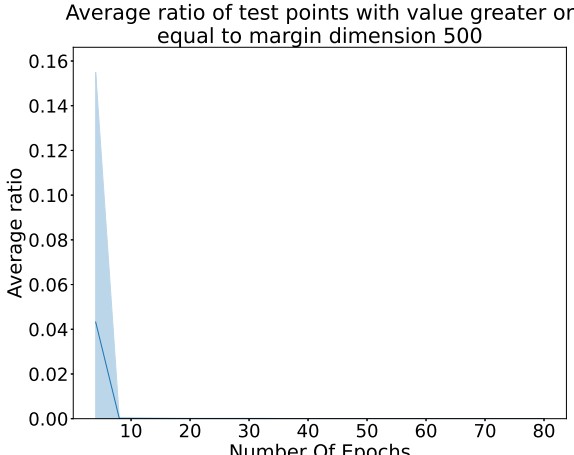

(a) The percentage of training points that lie on the margin (up to 10% slack) increases with the number of epochs.

(b) The percentage of test points that lie on or above the margin decreases with the number of epochs.

Figure 4: Effect of proximity to a KKT point on the proposed membership inference attack. As training progresses and the network approaches a KKT point, a larger fraction of training samples lie on the margin (a), while a smaller fraction of test samples lie on or above the margin (b). Results are averaged over 10 independent runs.

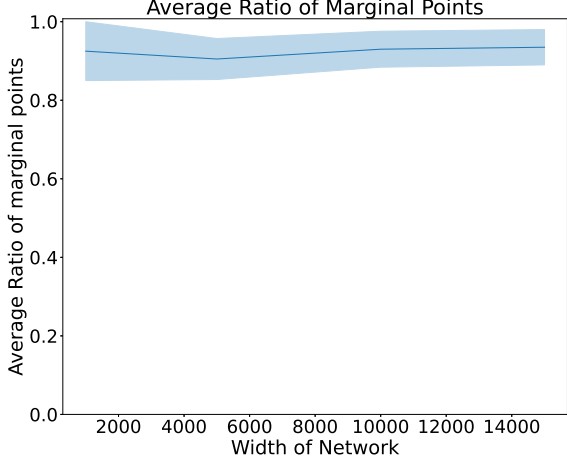

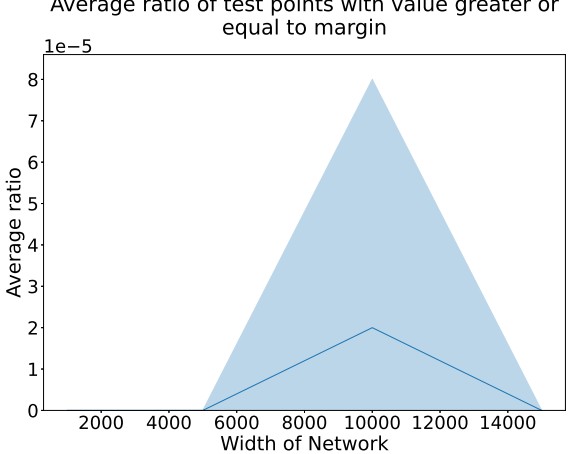

(a) The percentage of training points that lie on the margin (up to 10% slack) is independent of the network's width.

(b) The percentage of test points that lie on or above the margin is independent of the network's width. Y-axis is of scale $10^{-5}$.

Figure 5: Effect of network width on the proposed membership inference attack. As the width of the hidden layer increases, the proportion of training samples lying on the margin (a) and the proportion of test samples lying on or above the margin (b) remain essentially unchanged. These results support the theoretical analysis. Results are averaged over 10 independent runs.

## F    Proofs for Subsection 4.2

**Lemma F.1.** *Let $\Phi$ be a 2-layer homogeneous network that satisfy the KKT conditions. Let $x_l < x_{l+1}$ be 2 adjacent marginal training points. The number of breapoints in the interval $(x_l, x_{l+1})$ is at most 2, i.e.*

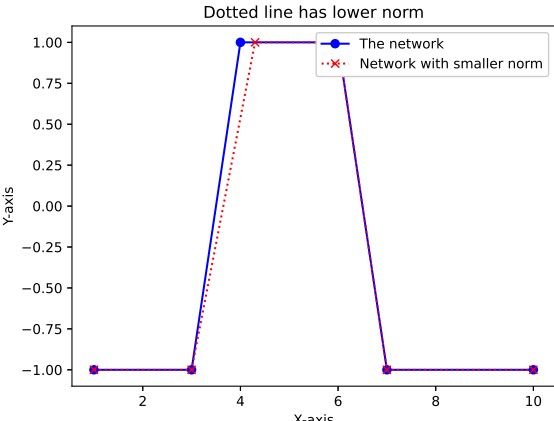

Figure 6: The blue network is a network which the breapoint is not a training point. The dotted-red network has smaller norm.

$|\{-\frac{b_j}{w_j} : \; x_l < -\frac{b_j}{w_j} < x_{l+1}\}| \leq 2$. *Moreover, if there are 2 breapoints, the neurons that form the breapoints must have different signs.*

*Proof.* Let $c_{j_1}(x) = v_{j_1}[w_{j_1}x + b_{j_1}]_+$ and $c_{j_2}(x) = v_{j_2}[w_{j_2}x + b_{j_2}]_+$ be 2 neurons with $w_{j_1} < 0$ and $w_{j_2} < 0$ such that their breapoints are between $x_l$ and $x_{l+1}$. Both $c_{j_1}$ and $c_{j_2}$ are determined by all training points that are smaller than $x_{l+1}$. Let us examine their breapoint $-\frac{b_l}{w_l}$ and $-\frac{b_{l+1}}{w_{l+1}}$: From Eq. (8) and Eq. (9) we get that

$$-\frac{b_{j_1}}{w_{j_1}} = -\frac{v_{j_1}\sum_{i=1}^l \lambda_i y_i}{v_{j_1}\sum_{i=1}^l \lambda_i y_i x_i} = -\frac{\sum_{i=1}^l \lambda_i y_i}{\sum_{i=1}^l \lambda_i y_i x_i} = -\frac{v_{j_2}\sum_{i=1}^l \lambda_i y_i}{v_{j_2}\sum_{i=1}^l \lambda_i y_i x_i} = -\frac{b_{j_2}}{w_{j_2}}$$

This means the neurons have the same breapoint and are active on the same region, which means they are the same neuron.

The same argument can be made to show that if $w_l > 0$ and $w_{l+1} > 0$ the neurons have the same breapoint. We conclude that in this interval we can have at most one neuron with $w > 0$ and at most one neuron with $w < 0$ with breapoints in the interval $[x_l, x_{l+1}]$. □

**Lemma F.2.** *Let $x_1 < x_2 < \cdots < x_n$ be the training points on the margin and $\Phi(x; \theta)$ be a 2-layers NN. If The network $\Phi(x; \theta)$ satisfies the KKT conditions and is not constant in any interval; then the number of times it crosses the margin is at most $6n$.*

*Proof.* between each $x_l$, $x_{l+1}$ there are at most 2 breapoints, i.e. the networks cross the margin at most 6 times in the interval $[x_l, x_{l+1}]$ (3 times the margin $y = 1$ and 3 times the margin $y = -1$). Before the point $x_1$ and after the point $x_n$, the network crosses the line at most 6 times in each interval. So, if we sum up all the crosses, we see that the network crosses the margin at most $6 \cdot (n-2) + 12 = 6n$ □

*Proof of Thm. 4.2.* Assume towards contradiction that there are no training points in the interval $[-\frac{b_{i-1}}{w_{i-1}}, -\frac{b_{i+1}}{w_{i+1}}]$. Since there are 3 breapoints, two of the neurons must have the same sign. Assume w.l.o.g. that $sgn(w_{i-1}) = sgn(w_i)$ (all other cases are similar). Since there are no marginal training data in $[-\frac{b_{i-1}}{w_{i-1}}, -\frac{b_{i+1}}{w_{i+1}}]$, they are active on the same set of training points, which means by Eq. (8) and Eq. (9) that $-\frac{b_{i-1}}{w_{i-1}} = -\frac{b_i}{w_i}$.

Each interval crosses the margin at most twice, so the number points lying on the margin is at most 4. □

*Proof of Thm. 4.3.* This proof follows the same logic as the proof of Lemma A.6 in Kornowski et al. (2023).

Assume towards contradiction that neither $-\frac{b_i}{w_i}$ nor $-\frac{b_{i+1}}{w_{i+1}}$ are in the training set, if $x \in [-\frac{b_i}{w_i}, -\frac{b_{i+1}}{w_{i+1}}]$ then $x \in (-\frac{b_i}{w_i}, -\frac{b_{i+1}}{w_{i+1}})$.

Note that $sgn(w_{i-1}) = -sgn(w_i)$ because there is no training point in the interval $(-\frac{b_i}{w_i}, -\frac{b_{i+1}}{w_{i+1}})$ so by F.1 they must have different signs.

Also note that there must be a training point either in $[-\frac{b_{i-2}}{w_{i-2}}, -\frac{b_{i-1}}{w_{i-1}}]$ or in $[-\frac{b_i}{w_i}, -\frac{b_{i+1}}{w_{i+1}}]$ (or in both). If it is not the case there are at least 3 breapoints between to training data points, contradiction to F.1.

**CASE 1**: $v_i^2 + \frac{v_i w_i v_{i-1}}{w_{i-1}} + \frac{b_i}{1-\delta}(\frac{w_i b_{i-1}}{w_{i-1}} - b_i) - \frac{w_1 v_i w_i}{v_1} - \frac{b_1 b_{i-1} v_i w_i}{v_1 w_{i-1}} > 0$
Define the following neural network:

$$
\begin{aligned}
\Phi(\boldsymbol{\theta}_\delta; x) := &\sum_{j \in [n] \setminus \{i-1, i, 1\}} v_j \left[w_j \cdot x + b_j\right]_+ + \left(1 - \delta \frac{v_i w_i}{v_{i-1} w_{i-1}}\right) v_{i-1} \left[w_{i-1} x + b_{i-1}\right]_+ + \\
&(1-\delta) v_i \left[w_i x + b_i - \frac{\delta}{1-\delta}\left(\frac{w_i b_{i-1}}{w_{i-1}} - b_i\right)\right]_+ + \\
&v_1 \left[\left(w_1 + \delta \frac{v_i w_i}{v_1}\right) x + \left(b_1 + \delta \frac{v_i w_i b_{i-1}}{v_1 w_{i-1}}\right)\right]_+
\end{aligned}
$$

For small enough $\delta$, the new breapoints do not cross any training point so for any training point $x_j$ we have that $\Phi(\boldsymbol{\theta}; x_j) = \Phi(\boldsymbol{\theta}_\delta, x_j)$ and in particular $\Phi(\boldsymbol{\theta}_\delta, x)$ satisfies the margin condition for each training point $x_j$. Also note that $\|\Phi(\boldsymbol{\theta}; x) - \Phi(\boldsymbol{\theta}_\delta; x)\|^2 \to 0$ as $\delta \to 0$. Let us compute $\|\Phi(\boldsymbol{\theta}_\delta, x)\|^2$:

$$
\begin{aligned}
\|\Phi(\boldsymbol{\theta}_\delta; x)\|^2 = &\sum_{j \in [n] \setminus \{i-1, i, 1\}} (v_j^2 + w_j^2 + b_j^2) + \left(1 - \delta \frac{v_i w_i}{v_{i-1} w_{i-1}}\right)^2 v_{i-1}^2 + w_{i-1}^2 + b_{i-1}^2 + \\
&(1-\delta)^2 v_i^2 + w_i^2 + \left(b_i - \frac{\delta}{1-\delta}\left(\frac{w_i b_{i-1}}{w_{i-1}} - b_i\right)\right)^2 + \\
&v_1^2 + \left(w_1 + \delta \frac{v_i w_i}{v_1}\right)^2 + \left(b_1 + \delta \frac{v_i w_i b_{i-1}}{v_1 w_{i-1}}\right)^2 = \\
&\|\Phi(\boldsymbol{\theta}; x)\|^2 - 2\delta\left(v_i^2 + \frac{v_i w_i v_{i-1}}{w_{i-1}} + \frac{b_i}{1-\delta}(\frac{w_i b_{i-1}}{w_{i-1}} - b_i) - \frac{w_1 v_i w_i}{v_1} - \frac{b_1 b_{i-1} v_i w_i}{v_1 w_{i-1}}\right) + O(\delta^2) \\
&< \|\Phi(\boldsymbol{\theta}; x)\|^2
\end{aligned}
$$

**CASE 2**: $v_i^2 + \frac{v_i w_i v_{i-1}}{w_{i-1}} + \frac{b_i}{1-\delta}(\frac{w_i b_{i-1}}{w_{i-1}} - b_i) - \frac{w_1 v_i w_i}{v_1} - \frac{b_1 b_{i-1} v_i w_i}{v_1 w_{i-1}} < 0$
Define the following neural network:

$$
\begin{aligned}
\Phi(\boldsymbol{\theta}_\delta; x) := &\sum_{j \in [n] \setminus \{i-1, i, 1\}} v_j \left[w_j \cdot x + b_j\right]_+ + \left(1 + \delta \frac{v_i w_i}{v_{i-1} w_{i-1}}\right) v_{i-1} \left[w_{i-1} x + b_{i-1}\right]_+ + \\
&(1+\delta) v_i \left[w_i x + b_i + \frac{\delta}{1+\delta}\left(\frac{w_i b_{i-1}}{w_{i-1}} - b_i\right)\right]_+ + \\
&v_1 \left[\left(w_1 - \delta \frac{v_i w_i}{v_1}\right) x + \left(b_1 - \delta \frac{v_i w_i b_{i-1}}{v_1 w_{i-1}}\right)\right]_+
\end{aligned}
$$

The norm $\|\Phi(\boldsymbol{\theta}_\delta; x)\|^2$ is:

$$\|\Phi(\boldsymbol{\theta}_\delta; x)\|^2 = \sum_{j \in [n] \setminus \{i-1, i, 1\}} (v_j^2 + w_j^2 + b_j^2) + \left(1 + \delta \frac{v_i w_i}{v_{i-1} w_{i-1}}\right)^2 v_{i-1}^2 + w_{i-1}^2 + b_{i-1}^2 +$$

$$(1+\delta)^2 v_i^2 + w_i^2 + \left(b_i + \frac{\delta}{1-\delta}(\frac{w_i b_{i-1}}{w_{i-1}} - b_i)\right)^2 +$$

$$v_1^2 + \left(w_1 - \delta \frac{v_i w_i}{v_1}\right)^2 + \left(b_1 - \delta \frac{v_i w_i b_{i-1}}{v_1 w_{i-1}}\right)^2 =$$

$$\|\Phi(\boldsymbol{\theta}; x)\|^2 - 2\delta \left(-v_i^2 - \frac{v_i w_i v_{i-1}}{w_{i-1}} - \frac{b_i}{1-\delta}(\frac{w_i b_{i-1}}{w_{i-1}} - b_i) + \frac{w_1 v_i w_i}{v_1} + \frac{b_1 b_{i-1} v_i w_i}{v_1 w_{i-1}}\right) + O(\delta^2)$$

$$< \|\Phi(\boldsymbol{\theta}; x)\|^2$$

**CASE 3**: $v_i^2 + \frac{v_i w_i v_{i-1}}{w_{i-1}} + \frac{b_i}{1-\delta}(\frac{w_i b_{i-1}}{w_{i-1}} - b_i) - \frac{w_1 v_i w_i}{v_1} - \frac{b_1 b_{i-1} v_i w_i}{v_1 w_{i-1}} = 0$
In this case, define the following neural network:

$$\Phi(\boldsymbol{\theta}_\delta; x) := \sum_{j \in [n] \setminus \{i-1, i, 1\}} v_j \left[w_j \cdot x + b_j\right]_+ +$$

$$(1-\delta) v_{i-1} \left[w_{i-1} x + b_{i-1} - \frac{\delta}{1-\delta}\left(\frac{w_{i-1} b_i}{w_i} - b_{i-1}\right)\right]_+ +$$

$$\left(1 - \delta \frac{v_{i-1} w_{i-1}}{v_i w_i}\right) v_i \left[w_i x + b_i\right]_+ +$$

$$v_1 \left[\left(w_1 + \delta \frac{v_{i-1} w_{i-1}}{v_1}\right) x + b_1 + \delta \frac{v_{i-1} w_{i-1} b_i}{v_1 w_i}\right]_+$$

Before computing the norm, note two observations:

1. By assumption, $v_i w_i = -v_{i-1} w_{i-1}$ and hence $\frac{v_i}{w_{i-1}} = -\frac{v_{i-1}}{w_i}$,

2. By definition of case 3, $v_i^2 + \frac{v_i w_i v_{i-1}}{w_{i-1}} + \frac{b_i}{1-\delta}\left(\frac{w_i b_{i-1}}{w_{i-1}} - b_i\right) - \frac{w_1 v_i w_i}{v_1} - \frac{b_1 b_{i-1} v_i w_i}{v_1 w_{i-1}} = 0$,

Now let us compute the norm:

$$\|\Phi(\boldsymbol{\theta}_\delta; x)\|^2 = \sum_{j \in [n] \setminus \{i-1, i, 1\}} (v_j^2 + w_j^2 + b_j^2) + (1-\delta)^2 v_{i-1}^2 + w_{i-1}^2 + \left(b_{i-1} - \frac{\delta}{1-\delta}(\frac{w_{i-1} b_i}{w_i} - b_{i-1})\right)^2 +$$

$$\left(1 - \delta \frac{v_{i-1} w_{i-1}}{v_i w_i}\right)^2 v_i^2 + w_i^2 + b_i^2 +$$

$$\left(w_1 + \delta \frac{v_{i-1} w_{i-1}}{v_1}\right)^2 + \left(b_1 + \delta \frac{v_{i-1} w_{i-1} b_i}{v_1 w_i}\right)^2 =$$

$$\|\Phi(\boldsymbol{\theta}; x)\|^2 - 2\delta \left(v_{i-1}^2 + \frac{b_{i-1}}{1-\delta}(\frac{w_{i-1} b_i}{w_i} - b_{i-1}) + \frac{v_{i-1} w_{i-1} v_i}{w_i} - \frac{w_1 v_{i-1} w_{i-1}}{v_1} - \frac{b_1 b_i v_{i-1} w_{i-1}}{v_1 w_i}\right) + O(\delta^2)$$

We need to show that

$$v_{i-1}^2 + \frac{b_{i-1}}{1-\delta}\left(\frac{w_{i-1} b_i}{w_i} - b_{i-1}\right) + \frac{v_{i-1} w_{i-1} v_i}{w_i} - \frac{w_1 v_{i-1} w_{i-1}}{v_1} - \frac{b_1 b_i v_{i-1} w_{i-1}}{v_1 w_i} \neq 0$$

(if $v_{i-1}^2 + \frac{b_{i-1}}{1-\delta}(\frac{w_{i-1} b_i}{w_i} - b_{i-1}) + \frac{v_{i-1} w_{i-1} v_i}{w_i} - \frac{w_1 v_{i-1} w_{i-1}}{v_1} - \frac{b_1 b_i v_{i-1} w_{i-1}}{v_1 w_i} < 0$ then, as in the previous cases, we change every $\delta$ to $-\delta$ and every $-\delta$ to $\delta$).

By observation 1 we know that:

$$\frac{v_{i-1}w_{i-1}v_i}{w_i} = -\frac{v_i w_i v_i}{w_i} = -v_i^2 \tag{14}$$

$$\frac{v_i w_i v_{i-1}}{w_{i-1}} = -\frac{v_{i-1}w_{i-1}v_i}{w_{i-1}} = -v_{i-1}^2 \tag{15}$$

Combine this with observation 2 we get:

$$v_i^2 + \frac{v_i w_i v_{i-1}}{w_{i-1}} + \frac{b_i}{1-\delta}\left(\frac{w_i b_{i-1}}{w_{i-1}} - b_i\right) - \frac{w_1 v_i w_i}{v_1} - \frac{b_1 b_{i-1} v_i w_i}{v_1 w_{i-1}} = 0$$

$$\Rightarrow \frac{b_i}{1-\delta}\left(\frac{w_i b_{i-1}}{w_{i-1}} - b_i\right) - \frac{b_1 b_{i-1} v_i w_i}{v_1 w_{i-1}} = \frac{w_1 v_i w_i}{v_1} - \frac{v_i w_i v_{i-1}}{w_{i-1}} - v_i^2$$

$$\Rightarrow \frac{b_i}{1-\delta}\left(\frac{w_i b_{i-1}}{w_{i-1}} - b_i\right) - \frac{b_1 b_{i-1} v_i w_i}{v_1 w_{i-1}} = v_{i-1}^2 - v_i^2 + \frac{w_1 v_i w_i}{v_1}, \tag{16}$$

where Eq. (16) follows by substitution of Eq. (15). Rewriting the equation in case 3 using Eq. (14), we need to show that

$$v_{i-1}^2 - v_i^2 + \frac{w_1 v_i w_i}{v_1} + \frac{b_{i-1}}{1-\delta}\left(\frac{w_{i-1}b_i}{w_i} - b_{i-1}\right) - \frac{b_1 b_i v_{i-1} w_{i-1}}{v_1 w_i} \neq 0$$

and using Eq. (16), we can further simplify it to

$$\frac{b_i}{1-\delta}\left(\frac{w_i b_{i-1}}{w_{i-1}} - b_i\right) - \frac{b_1 b_{i-1} v_i w_i}{v_1 w_{i-1}} + \frac{b_{i-1}}{1-\delta}\left(\frac{w_{i-1}b_i}{w_i} - b_{i-1}\right) + \frac{b_1 b_i v_i w_i}{v_1 w_i} \neq 0$$

That expression can be rewritten as

$$\frac{b_1 v_i(b_i w_{i-1} - b_{i-1} w_i)}{v_1 w_{i-1}} + \frac{1}{1-\delta}\left(-b_{i-1}^2 + \frac{b_{i-1}b_i w_i}{w_{i-1}} + \frac{b_{i-1}b_i w_{i-1}}{w_i} - b_i^2\right)$$

The only way this expression is equal to 0 for every sufficiently small $\delta > 0$ is when both summands are 0. let us look at the second summand.

$$\frac{1}{1-\delta}\left(-b_{i-1}^2 + \frac{b_{i-1}b_i w_i}{w_{i-1}} + \frac{b_{i-1}b_i w_{i-1}}{w_i} - b_i^2\right) = \frac{1}{1-\delta}\left(-b_{i-1}^2 + b_{i-1}b_i\left(\frac{w_i}{w_{i-1}} + \frac{w_{i-1}}{w_i}\right) - b_i^2\right) \leq$$

$$\frac{1}{1-\delta}\left(-b_{i-1}^2 - 2b_{i-1}b_i - b_i^2\right) = -\frac{1}{1-\delta}(b_{i-1} + b_i)^2$$

Where the inequality stems from the inequality $x + \frac{1}{x} \leq -2$ for every $x < 0$ (and equality holds when $x = -1$) where in our case $x = \frac{w_i}{w_{i-1}}$, and they have different signs so $\frac{w_i}{w_{i-1}} < 0$. For the summand to be 0 it must holds that $w_i = -w_{i-1}$ and $b_i = -b_{i-1}$, but that can not happen because if that would have happened then $-\frac{b_i}{w_i} = -\frac{b_{i-1}}{w_{i-1}}$; i.e., the two neurons have the same breakpoint.

An example of a network with smaller norm can be found in Figure 6. $\qquad \square$

*Proof of Thm. 4.4.* We prove that for each iteration, we add at least one training point to the set $S$. As the number of iteration is finite, and in each iteration the number of points added to $S$ are finite, $S$ is finite.

If the condition in line 6 in Algorithm 1 is met, by Thm. 4.2 one of the points added to $S$ must be a training point, and the number of such points is at most 4.

If both conditions at lines 9 and 11 are met, by Thm. 4.3 either $y$ or $z$ is a training point. So the ratio of training points in $S$ is at least $\frac{1}{4}$ $\qquad \square$

## G  Data reconstruction without a local minimum assumption

The following is an alternative algorithm to Algorithm 1, which performs data reconstruction while relaxing the assumption that the network has converged to a local minimum of the max margin problem. The

algorithm also relaxes the assumption that there is a neuron which is active on all the dataset. The difference here is that instead of alternating between Thm. 4.3 and Thm. 4.4, the following algorithm only utilizes Thm. 4.3. While this limits its applicability, it allows us to invoke it with weaker assumptions, given that the structure of the network we are attacking allows us to do so by having intervals that do not lie on the margin.

**Theorem G.1.** *Let $\Phi : \mathbb{R} \to \mathbb{R}$ be a 2-layer homogeneous network that satisfies the KKT conditions, then the following algorithm builds a finite set of which a constant ratio $p \geq \frac{1}{4}$ of the points are training points. In words, the algorithm iterates over the intervals of the network that are not constant on the margin, and*

---

**Algorithm 2:** Build a finite set of candidates

---

$S \leftarrow \emptyset$
**for** $i \leftarrow 1$ **to** $k-2$ **do**
  $\quad x \leftarrow -\frac{b_i}{w_1}$
  $\quad y \leftarrow -\frac{b_{i+1}}{w_{i+1}}$
  $\quad z \leftarrow -\frac{b_{i+2}}{w_{i+2}}$
  $\quad$ **if** *both $[x, y]$ and $[y, z]$ do not lie on the margin* **then**
    $\quad\quad S \leftarrow S \cup \{p : p \in [x, y] \wedge p \text{ is on the margin}\}$
    $\quad\quad S \leftarrow S \cup \{p : p \in [y, z] \wedge p \text{ is on the margin}\}$

---

*adds to the set of candidates the points that lie on the margin.*

*Proof of Theorem Thm. G.1.* The analysis is very similar to the proof of Algorithm 1. In each iteration, by Thm. 4.2 one of the points added to $S$ must be a training point, and the number of such points is at most 4, so the fraction of training points in $S$ is at least $\frac{1}{4}$.

$\square$

We remark that the above algorithm is always applicable to some extent, except for the extreme case where the network alternates between a constant and non-constant intervals (for which we have Algorithm 1). This demonstrates that the assumption that the network has converged to a local minimum is not necessary for performing our reconstruction attack in all except for extreme instances.

