# OpenReview forum: "Provable Privacy Attacks on Trained Shallow Neural Networks"
_TMLR — Under review for TMLR_

### Review · Reviewer_QzxG · 2026-06-25

**Summary Of Contributions:**

The paper provides a theoretical assessment of training-set membership and training-set reconstruction attacks on 2-layer Relu networks. The paper is motivated by an earlier work, which developed an attack for homogeneous networks and demonstrated the effectiveness of that attack empirically. The core observation is the observation that homogeneous networks trained with logistic-like loss functions converge to KKT points of the so-called maximum margin problem. This allows interpreting the learned parameters as a solution to a Lagrangian like constraint problem, with particular input-output constraints for the training set. The attack uses this relation to construct a learning task, whose purpose is to fully reconstruct the training set, by utilizing the constraint characterization to design a loss function. A more thorough analysis of this method in a subsequent work however unveiled that solutions to this attack-learning task can differ arbitrarily from the training set, questioning the general applicability of the attack. The present paper reconsiders the principle observation behind this attack, but instead makes use of the margin value given by the KKT point in order to classify an input as being or not being in the training set. The authors show that this attack provably works under specific assumptions on the data and training set distributions and on the neural network. The core observation is that points in the training set will receive the margin value as prediction, whereas points outside the training set will be assigned significantly smaller (absolute) values. Besides using this result for determining training-set membership, the authors further expand it to design a provably correct training-set reconstruction algorithm for univariate networks. The validity and impact of the assumptions along with the main theoretical results are empirically evaluated using a controlled and purely synthetic data and learning regime.

**Audience:**

No

**Audience Explanation:**

Unfortunately, I need to supplement the previous point with a big BUT. In principle, this work has the potential of attracting broad interest. However, the assumptions that are being made are extremely restrictive. Going beyond a pure mathematical exercise, the relevance of this work is not completely unclear. In the end, it is not obvious who is going to be interested in or benefit from the presented analysis. Four assumptions stand out particularly: (1) the paper focuses exclusively on 2-layer neural networks with ReLU activations and limiting bias terms to the first layer, (2) the attacks require tiny training sets, tiny to the point making it completely unclear what one should learn from so few examples, (3) the results assume a converged network (a KKT point), and (4) the univariate case has essentially no practical relevance. Like said, I totally acknowledge that a theoretical analysis requires making assumptions. However, to the least, I would then expect seeing a proper justification of why the particular assumptions need to be made. At presence, it is completely open whether the shown properties could be maintained when relaxing some of the assumptions, or what difficulties arise when doing so. Moreover, I would assume some kind of discussion of possible ways to relax the assumptions in the future. Neither of the points are addressed in the paper. Currently, it looks like the authors did some "simple" math calculations to obtain the desired results, and extracted the assumptions needed for the transformation to be correct. All in all, I value the results -- at least in the way they are currently presented -- to be of limited interest to researchers, especially given that the underlying idea of the attack (Thm 2.1) is not new and was published before.

**Broader Impact Concerns:**

I don't see any ethical concerns.

**Claims And Evidence:**

Yes

**Claims Explanation:**

Attacks recovering potentially sensible information from the training set are of high interest these days. The paper tackles the search for such methods from a theoretical angle, contributing formal conditions and insights under which such attacks are successful. Naturally for complex theoretical problems, simplifying assumptions need to be made. The paper introduces clearly and formally the particular assumptions under which the provided theoretical properties hold. The proofs of the claims are thorough, and to the degree I could verify them, seem to be correct. With this respect, the claims made by the authors seem to be well-supported.

**Requested Changes:**

The most important point for improving the present paper is working out the assumptions. Me as a reader, I want to be convinced of why a) the assumptions are necessary, and b) why they make sense, i.e., why they are reasonable in practice. In particular, the authors should make clear
- Why one particularly focuses on 2-layer NNs. Thm 2.1 should hold for any homogeneous NN, so in particular NNs with deeper architectures (albeit still limiting non-zero bias terms to the very first layer). Where is the assumption of having only 2 layers necessary? What breaks if we would consider deeper NNs? Is there reason to believe that the results can be extended beyond this restrictive case?
- Assumption 3.1 needs to be better explained and the impact needs to be justified. Assumption 3.1 on its own reads rather abstract, and the paper is currently missing a brief description of what the conditions express exactly. Moreover, a brief lookahead should be provided that briefly explains why those assumptions need to be made (what breaks if we don't have them). Crucially, currently the assumptions seems to enforce that the size of training set is absolutely tiny compared to the feature dimension. It is kind of obvious that a highly over-parameterized NNs will remember by heart tiny training data sets, which significantly weakens the theoretical results. The authors should provide a convincing reason of why this assumption is justified, and why the theoretical results are noteworthy despite these restrictions. This limitation is especially visible in the empirical demonstration of some of the claims, where NNs are trained with just a handful of instances. I cannot imagine any setting where a useful model can be learned with so few training data.
- The paper makes the very strong assumption of having a fully converged NN. In practice, we will never have such a function. A theoretical attack analysis would hence make much more sense considering approximately converged NNs. Previous works have introduced such notions. The authors should point to this aspect, and argumentatively highlight in how far their theoretical results could be adapted to this approximate property.
- In general, pertaining to all the assumptions, the authors should show a perspective as to how these assumptions could potentially be addressed in the future, or they should provide at least a good explanation which difficulties arise when trying to relax them.

Finally, the theoretical claims should be made more precise. Almost all the claims have the structure: case 1) for points in the training set; case 2) for points in the data distribution. Under the typical assumption that the training set is itself sampled from the data distribution, i.e., the training set is a proxy of the data distribution, this distinction does in my view not really make sense. In particular, the two cases are not disjoint.

---

> ### Author Response · Authors · 2026-07-14
>
> We thank the reviewer for the detailed review. We address the comments and questions raised by the reviewer below.
>
> - Two-layer networks: our restriction to two-layer networks is driven by a genuine technical barrier rather than by a belief that the underlying phenomenon disappears with depth. For a two-layer ReLU network, the KKT conditions give an explicit representation of each first-layer weight as a weighted combination of the training instances. This representation is the key step that allows us to express the prediction on a query point in terms of its inner products with the training instances and thereby exploit near orthogonality. For deeper networks, the KKT equations couple multiple layers through products of weights and data-dependent activation patterns. Consequently, they no longer yield the direct representation used in our proof, and near orthogonality alone does not provide sufficient control over the resulting expressions. Thus, our argument does not show that the attack fails for deeper networks; rather, the available KKT characterization becomes substantially harder to translate into a meaningful privacy guarantee. This difficulty is reflected in the theoretical literature, where comparable KKT-based analyses overwhelmingly focus on two-layer networks [1-6, 8-11].
> More generally, a significant portion of the deep-learning-theory literature considers shallow neural networks as a testbed for understanding complex phenomena, so our approach here is quite standard.
> To investigate whether the phenomenon itself extends beyond our proof, we empirically tested the behavior of deeper architectures in Appendix E. These experiments are not intended to establish or validate a theorem for deeper architectures, but they indicate that the predicted behavior may persist beyond the setting currently covered by our analysis. Extending the theoretical argument to deeper networks remains an important open direction.
>
> - Assumption 3.1: We agree that the regime $d \gg n^2$ should not be presented as necessarily realistic, but rather as a standard and analytically tractable setting for high-dimensional theory, similar to assumptions used in prior work [1–6]. We also provide several standard distributions satisfying the assumption, which have been considered in related theoretical analyses [1,5,7]. Some form of separation is necessary for our output-based membership-inference attack: since $\Phi$ is Lipschitz, sufficiently close points receive similar predictions and may be difficult to distinguish. Near-orthogonality is a convenient sufficient condition that enables us to prove a clear separation between training and fresh samples. The scaling $d \gg n^2$ is likewise a sufficient condition ensuring that this separation holds uniformly with high probability, rather than a necessary threshold. Our experiments in Appendix E, including experiments on MNIST and in substantially more moderate dimensions, indicate that the predicted behavior can persist even when the formal scaling assumptions are violated. Finally, Appendix C shows that this regime is compatible with statistically meaningful learning and is not limited to arbitrary memorization.
>
> - Convergence to an exact KKT point: This assumption is also standard in related work [1–3]. Since, to the best of our knowledge, this is the first theoretical study of the privacy implications of this implicit bias, we initially focused on the analytically tractable exact-KKT setting. Following the reviewer’s suggestion, we added Section~5, titled "Beyond Exact KKT," which explains how our guarantees extend to approximate KKT points.
>
> - Univariate data: While restrictive from a practical viewpoint, this assumption is commonly used in theoretical papers [8-10]. Moreover, the work of [11] suggests that reconstruction attacks in high dimensions are much more difficult to execute, which motivates the study of a complementary, lower-dimensional setting.
>
> - Relaxing the assumptions: Following the reviewer’s suggestion, we added discussions throughout the paper explaining how each assumption might be relaxed and which technical obstacles arise. We also revised the presentation to more clearly highlight the experiments already included in Appendix E, which examine deeper architectures, more moderate dimensions, and realistic data, and probe whether the qualitative behavior predicted by our theory persists beyond the regime covered by our guarantees.
>
> - Theoretical claims should be made more precise: We thank the reviewer for pointing out this potential ambiguity. Our two cases correspond to different sampling procedures: a member is selected from the realized training set, whereas a fresh point is sampled independently from the underlying distribution after the training set and model have been fixed. Although both points have the same marginal distribution, only the former is statistically dependent on the trained model. We added a few sentences clarifying this convention.

---

> > ### Author Response · Authors · 2026-07-14
> >
> > References:
> >
> > [1] Gal Vardi, Gilad Yehudai, and Ohad Shamir. Gradient methods provably
> > converge to non-robust networks.
> >
> > [2] Spencer Frei, Gal Vardi, Peter Bartlett, and Nathan Srebro. Benign
> > overfitting in linear classifiers and leaky relu networks from KKT conditions for
> > margin maximization.
> >
> > [3] Spencer Frei, Gal Vardi, Peter L. Bartlett AND Nathan Srebro. The Double-Edged Sword of Implicit Bias: Generalization vs. Robustness in ReLU Networks.
> >
> > [4] Spencer Frei and Gal Vardi and Peter L. Bartlett and Nathan Srebro and
> > Wei Hu. Implicit Bias in Leaky ReLU Networks Trained on High-Dimensional
> > Data.
> >
> > [5] Guy Kornowski, Gilad Yehudai, and Ohad Shamir. From tempered to
> > benign overfitting in relu neural networks.
> >
> > [6] Yiwen Kou, Zixiang Chen and Quanquan Gu. Implicit Bias of Gradient Descent for Two-layer ReLU and Leaky ReLU Networks on Nearly-orthogonal Data.
> >
> > [7] Noah Amsel and Gilad Yehudai and Joan Bruna. On the Benefits of
> > Rank in Attention Layers.
> >
> > [8] I. Safran, G. Vardi, and J. D. Lee. On the effective number of linear re-
> > gions in shallow univariate relu networks: Convergence guarantees and implicit
> > bias.
> >
> > [9] F. Williams, M. Trager, D. Panozzo, C. Silva, D. Zorin, and J. Bruna.
> > Gradient dynamics of shallow univariate relu networks.
> >
> > [10] J. Sahs, R. Pyle, A. Damaraju, J. O. Caro, O. Tavaslioglu, A. Lu, F.
> > Anselmi, and A. B. Patel. Shallow univariate relu networks as splines: Initial-
> > ization, loss surface, hessian, and gradient flow dynamics.
> >
> > [11] Yehonatan Refael, Guy Smorodinsky, Ofir Lindenbaum and Itay Safran. No Prior, No Leakage: Revisiting Reconstruction Attacks in Trained Neural Networks.
> >
> > [12] Noel Loo, Ramin Hasani, Mathias Lechner, Alexander Amini and Daniela Rus.
> > Understanding Reconstruction Attacks With The Neural Tangent Kernel And Dataset Distillation.

---

> > > ### Comment · Reviewer_QzxG · 2026-07-21
> > >
> > > Thanks for the clarifications. While I still believe that the assumptions make the results far off any practical relevance, I also completely understand their need in such a theoretical examination. With the changes in the abstract and the introduction, plus the additional discussions of the limitations throughout, I am reasonably satisfied with the revised paper.

---

### Review · Reviewer_G5bh · 2026-07-04

**Summary Of Contributions:**

The paper studies privacy leakage in trained two-layer ReLU networks. It assumes that training reaches a KKT point of a maximum-margin problem, then derives two privacy results. The main result is a membership inference attack in a high-dimensional near-orthogonal setting. The second result is a reconstruction claim for one-dimensional inputs, using the breakpoint structure of univariate ReLU networks.

**Strengths**

1. The paper addresses an interesting question.
It is useful to understand when membership inference and reconstruction attacks can be proved from the training dynamics of the learned model.

2. The high-dimensional membership inference result is conceptually clean.
Under the stated KKT and near-orthogonality assumptions, the paper gives a simple test based on output magnitude.
The separate cases for known margin, leaked point, and bounded margin also help explain what the attacker needs to know.

3. The additional experiments are useful.
The paper includes experiments for moderate dimension on MNIST with CNNs, approximate KKT behavior, and network width.
These experiments make the work more informative than a purely asymptotic theoretical analysis.

**Weaknesses**

1. Scope of the main claim.
The membership inference theorem is a sufficient-condition result under KKT and near-orthogonality assumptions. Some parts of the abstract and introduction read more broadly than this. The paper should state more directly that the theorem proves leakage in this specific regime.

2. Generalization in the experiments.
The experiments report margin separation, but they do not report training or test accuracy for the same models. This makes it hard to tell whether the signal appears in useful classifiers or mainly in small-sample memorization regimes.

3. Scope of the reconstruction result.
The reconstruction result is limited to one-dimensional inputs. It also relies on assumptions beyond Assumption 2.1, including a neuron active on all training points and local optimality. These conditions should be stated wherever the reconstruction contribution is summarized.

**Audience:**

Yes

**Audience Explanation:**

The paper gives a useful way to connect implicit bias and privacy leakage. Even with restrictive assumptions, this connection should interest some readers in privacy, membership inference, reconstruction attacks, and theory.

**Claims And Evidence:**

No

**Claims Explanation:**

The main theorems appear meaningful under their stated assumptions, and I did not identify a technical flaw in the main arguments I checked. My "No" reflects a gap between the wording of the claims and what is directly proved or measured.

**Requested Changes:**

**Major:**

1. I would like the authors to narrow the wording of the main claim.
In the abstract and introduction, it would help to state more explicitly that the membership inference result is proved under KKT and near-orthogonality assumptions.

2. It would help if training and test accuracy are reported for the experiments in Figure 1 and Appendix E.
These numbers would help determine whether the observed privacy signal appears in useful classifiers.

3. I suggest the reconstruction contribution to be stated more carefully.
The paper should make clear that the theorem is univariate and uses assumptions beyond Assumption 2.1.

**Optional:**

1. A figure for the univariate reconstruction argument would help. The current proof is easier to follow if the reader can see breakpoints, margin intervals, and candidate points.

---

> ### Author Response · Authors · 2026-07-14
>
> We thank the reviewer for the thoughtful and constructive feedback, for recognizing the significance of our theoretical contributions and experimental evaluation, and for the valuable suggestions on clarifying the scope of our results and strengthening the experimental presentation.
>
> Following the reviewer’s suggestions, we addressed the identified weaknesses and requested revisions as follows:
>
> - We narrowed the claims in the abstract and introduction by explicitly describing the membership-inference setting as one involving nearly orthogonal data.
>
> - We added training and test accuracies for all experiments. We clarified the scope of the reconstruction result, including an explicit statement of the additional assumptions required beyond convergence to a KKT point.

---

### Review · Reviewer_r5Ab · 2026-07-05

**Summary Of Contributions:**

This paper studies privacy leakage from the implicit bias of trained shallow ReLU networks. The main idea is to use existing KKT characterizations of homogeneous networks trained with logistic or exponential loss to derive privacy attacks.

For the high-dimensional setting, the paper shows that, under a near-orthogonality assumption on the data and assuming the trained two-layer network satisfies the relevant KKT conditions, training examples lie on the margin, while fresh examples from the same distribution tend to have much smaller outputs. This gives membership-inference attacks, depending on what the adversary knows about the margin.

The paper also has a one-dimensional reconstruction result. This part analyzes the breakpoints of univariate two-layer ReLU networks and gives an algorithm that returns a finite candidate set that contains the training points up to constant precision, under additional assumptions. There are also experiments suggesting that the membership-inference behavior may still show up outside the exact asymptotic assumptions used in the theory.

I think the main strength of the paper is that it studies an interesting question that has not been very well understood in theory: privacy leakage under the implicit-bias situation. The membership-inference result is clean and conceptually interesting. My main concern is that the reconstruction result is not always stated precisely enough, which makes it harder to understand exactly what is being guaranteed.

**Audience:**

Yes

**Audience Explanation:**

The paper studies a question that should be interesting to some parts of the TMLR audience, especially people working on privacy, memorization, and theory of neural networks.

I think the high-dimensional membership-inference result is the part that is likely to be of broadest interest. It gives a clean theoretical reason why training examples can have systematically larger outputs than fresh samples under the paper’s assumptions. The assumptions are restrictive, but the result still gives a useful mechanism and could motivate more work on when margin-based implicit bias creates privacy vulnerabilities.

The reconstruction result is more narrow. Still, it may be interesting to readers who care about the geometry of shallow ReLU networks, especially the relation between breakpoints, margins, and training data.

**Broader Impact Concerns:**

The paper studies privacy attacks, so there are clear dual-use concerns. On the positive side, the results can help us better understand when trained neural networks leak information, and this could be useful for designing better defenses or safer training procedures. On the negative side, the paper also gives theoretical guidance for how to use a trained model to infer membership or recover candidate training points.

I think the broader impact discussion should say this more explicitly. It should discuss responsible use, the limitations of the threat models, and why the results should mainly be used to evaluate and reduce privacy risks, rather than to attack deployed models.

**Claims And Evidence:**

Yes

**Claims Explanation:**

I think the main theoretical claim of the paper is mostly supported. The paper gives a plausible and interesting proof that trained two-layer ReLU networks can be vulnerable to membership inference under the implicit-bias and high-dimensional near-orthogonality assumptions. This is the core contribution of the paper, and I view it as the main value of the submission. The experiments are helpful as illustrations, but I do not think the paper relies on them.

My concerns are more about the precision of some claims than about whether there is a contribution. The novelty claim in the abstract, in particular “first provable vulnerabilities in this implicit-bias-driven setting,” seems reasonable as long as the qualifier “implicit-bias-driven” is kept consistently. However, some later statements, such as the claim that previous work is empirical, should be phrased carefully. Otherwise, it could be read as a broader claim about privacy attacks or membership inference in general.

The main issue for me is the reconstruction part. In several places, the paper seems to suggest that the method reconstructs a constant fraction of the training data. But the formal theorem and proof seem to show something weaker/different: the algorithm constructs a finite candidate set, and at least a constant fraction of the candidate points are training points. I think this distinction should be stated explicitly in the abstract, introduction, contribution list, and theorem discussion.

Finally, although the paper explains many of its assumptions, the reconstruction section would benefit from cleaner notation. For example, in Algorithm 1, the notation is confusing. Throughout the paper, n denotes the number of training samples and k denotes the number of neurons, but Algorithm 1 appears to iterate over neuron breakpoints -b_i/w_i with a loop bounded by n-2. If the loop is meant to be over breakpoints or neurons, the bound should use k, or the number of distinct breakpoint intervals. This part needs more clarification.

**Requested Changes:**

Requested changes:
1. Correct the informal statement of the univariate reconstruction result. In particular, the paper should distinguish between reconstructing a constant fraction of the training data and constructing a finite candidate set in which a constant fraction of the candidates are training points.
2. State all assumptions required for the reconstruction theorem whenever the result is summarized, including the assumptions beyond Assumption 2.1.
3. Clarify Algorithm 1 and make its indexing and loop bounds consistent with whether it is iterating over neurons, breakpoints, or breakpoint intervals.
4. Keep the novelty claim consistently scoped to the implicit-bias-driven setting.
5. Make the threat model and adversary knowledge easier to parse, ideally by summarizing the different attack settings in one place.
6. Clarify that the experiments are illustrative sanity checks beyond the exact theoretical setting, rather than evidence needed to validate the theorem under its stated assumptions.

---

> ### Author Response · Authors · 2026-07-14
>
> We thank the reviewer for their constructive review, and for their acknowledgment of the novelty of our work, especially the membership inference attack. We address their concerns and change requests below:
>
> 1. The informal statement of the univariate reconstruction result was corrected throughout as proposed to avoid misinterpretation.
>
> 2. Throughout the paper, we clarified and clearly stated the assumptions for the theorem that are missing.
>
> 3. We changed the indexing to be $k$ instead of $n$. In addition, we added a clarification explaining that we iterate over breakpoints, as the reviewer requested.
>
> 4. As requested, the scope was clarified wherever the novelty claim is discussed.
>
> 5. This is specified in the "Our contribution" subsection, where it is more accessible to the reader to compare the various assumptions made throughout our paper.
>
> 6. We added a clarification in the beginning of Section 3.2 that the experiments study whether our results hold under more general assumptions, and are not meant to validate our rigorous guarantees.
>
> Lastly, we added a broader impact discussion in the last paragraph of Section 6 as proposed.

---

> > ### Comment · Reviewer_r5Ab · 2026-07-17
> >
> > Thanks for addressing the requested changes in the revision. They look good to me.

---

### Author Response · Authors · 2026-07-14

We thank all reviewers for the detailed and comprehensive reviews. We addressed the requested changes in the revised version, and highlighted them in red for the reviewers' convenience. We would greatly appreciate any additional feedback on the changes made.